# Solution structures of multiple G-quadruplex complexes induced by a platinum(II)-based tripod reveal dynamic binding

Wenting Liu[1], Yi-Fang Zhong[1,2], Liu-Yi Liu[1], Chu-Tong Shen[1], Wenjuan Zeng[3], Fuyi Wang [3], Danzhou Yang[4] & Zong-Wan Mao[1,2]

DNA G-quadruplexes are not only attractive drug targets for cancer therapeutics, but also have important applications in supramolecular assembly. Here, we report a platinum(II)-based tripod (Pt-tripod) specifically binds the biological relevant hybrid-1 human telomeric G-quadruplex (Tel26), and strongly inhibits telomerase activity. Further investigations illustrate Pt-tripod induces the formation of monomeric and multimeric Pt-tripod–Tel26 complex structures in solution. We solve the 1:1 and the unique dimeric 4:2 Pt-tripod–Tel26 complex structures by NMR. The structures indicate preferential binding of Pt-tripod to the 5′-end of Tel26 at a low Pt-tripod/Tel26 ratio of 0–1.0. After adding more Pt-tripod, the Pt-tripod binds the 3′-end of Tel26, unexpectedly inducing a unique dimeric 4:2 structure interlocked by an A:A non-canonical pair at the 3′-end. Our structures provide a structural basis for understanding the dynamic binding of small molecules with G-quadruplex and DNA damage mechanisms, and insights into the recognition and assembly of higher-order G-quadruplexes.

[1] MOE Key Laboratory of Bioinorganic and Synthetic Chemistry, School of Chemistry, Sun Yat-Sen University, Guangzhou 510275, China. [2] College of Materials and Energy, South China Agricultural University, Guangzhou 510642, China. [3] Beijing National Laboratory for Molecular Sciences, National Centre for Mass Spectrometry in Beijing, CAS Key Laboratory of Analytical Chemistry for Living Biosystems, Institute of Chemistry, Chinese Academy of Sciences, Beijing 100190, China. [4] Department of Medicinal Chemistry and Molecular Pharmacology, Purdue University Center for Cancer Research, Purdue Institute for Drug Discovery, Purdue University, West Lafayette, IN 47907, USA. Correspondence and requests for materials should be addressed to D.Y. (email: yangdz@purdue.edu) or to Z.-W.M. (email: cesmzw@mail.sysu.edu.cn)

DNA G-quadruplexes have attracted increasing interest as potential drug targets for cancer therapy[1,2] and as important tools for supramolecular synthesis to build G-quadruplex-based nanomaterials with various functions[3,4]. G-quadruplexes are four-stranded structures formed in guanine-rich sequences that are held together by guanine–guanine Hoogsteen hydrogen bonding[5]. The formation of G-quadruplex structures is common and prevalent in regions with biological significance, such as human telomeres and oncogene promoter regions[6–8]. The human telomeric DNA, which consists of 5–8 kb tandem repeat sequences of $d(TTAGGG)_n$ and terminates with a 100–200 nt 3′ single-stranded overhang[9,10], caps by nucleoprotein complexes[11] and plays an important role in cell aging[12] and death[13]. Telomerase has been shown to be active in 80–85% of human cancers to extends the telomere and promote the lifespan of cancer cells[14,15]. Previous studies in human cancer cells have demonstrated DNA G-quadruplex structure formation in telomeres[16], and G-quadruplex stabilization by small molecules[17] induces tumour cell senescence and apoptosis by repressing telomerase activity[18] and the DNA damage response pathway[19,20]. Long telomeric-overhang DNA can form biologically relevant higher-order DNA structures containing consecutive G-quadruplexes[21–23] and provides specific binding sites for small molecules[24–26]. Thus, using small molecules to stabilize G-quadruplexes as potential drug targets for cancer therapy development has attracted great interest.

In recent decades, various small molecules have been developed to recognize, stabilize and probe biologically significant DNA G-quadruplexes[27–29]. To date, only a few solution complex structures of intramolecular G-quadruplexes targeted by small molecules have been determined[30–36]. However, all of them appear simple binding, that is, one or two molecules bound to one intramolecular G-quadruplex. In fact, ligand-induced higher-order G-quadruplex assembly structure was first reported by Hurley[37], but such structural details are challenging to obtain and still remain uncovered. Insights into the structural mechanism of how monomeric G-quadruplex assembles into higher-order G-quadruplexes induced and regulated by a small molecule are unknown. Structural information of higher-order G-quadruplex assemblies promoted by small molecules is essential for investigating the specific binding of biological relevant sequential G-quadruplexes in the long telomere.

Our lab previously synthesized and characterized a series of platinum(II) complexes that specifically bind and stabilize telomeric G-quadruplex structures with excellent anticancer properties[38–40]. Among these, a Pt-tripod (Fig. 1a) exhibited highly promising DNA-targeted photodynamic therapy anticancer activity both in vitro and in vivo[39]. The Pt-tripod is a three-fold-symmetric compound in a non-planar tertiary amine conformation. It possesses three pendant arms and each arm comprises two aromatic rings and a cationic platinum unit. Mechanistic investigation revealed that under light irradiation, the Pt-tripod rapidly damaged DNA, including G-quadruplex DNA. Here, we find that the Pt-tripod specifically binds to the biological relevant hybrid-1 human telomeric G-quadruplex DNA Tel26 (Fig. 1b) and strongly repress telomerase activity. We demonstrate the Pt-tripod induces the formation of multiple Pt-tripod–G-quadruplex complex structures, including monomeric, dimeric, and multimeric complex structures, by NMR, ESI-MS, native PAGE and FRET melting experiments. To understand the molecular mechanisms of multiple complexes induced by the Pt-tripod, solution NMR structures for the 1:1 and the dimeric 4:2 Pt-tripod–Tel26 complexes are solved and presented, revealing the dynamic binding and detailed interaction of the Pt-tripod.

## Results

**Specific binding of Tel26 G-quadruplex by Pt-tripod.** We used 1D $^1H$ NMR titration to investigate the interaction of the Pt-tripod and various G-quadruplex DNA sequences (Fig. 1c, Supplementary Fig. 1–4 and Supplementary Table 1). The results showed that Pt-tripod bound the intramolecular hybrid-1 human telomeric G-quadruplex Tel26 (Fig. 1b) with high specificity. In the $^1H$ NMR titration spectra (Fig. 1c), as Pt-tripod was gradually titrated into a Tel26 solution, a different set of distinct imino peaks appeared at a 0.5:1 Pt-tripod/Tel26 ratio. Signals for free Tel26 were still observed, indicating that Pt-tripod binding to Tel26 has a slow exchange rate on the NMR time-scale. At a 1:1 ratio, only 12 well-resolved peaks from the 1:1 Pt-tripod–Tel26 complex were observed. Interestingly, at a 2:1 ratio, another set of 13 imino peaks appeared, exhibiting slow exchange with the 1:1 complex. And only peaks for the second Pt-tripod–Tel26 complex were observed at a 3:1 ratio, revealing the presence of a single dominating complex structure. At a higher 4:1 ratio, the $^1H$ NMR spectrum broadened greatly. Notably, the Tel26 G-quadruplex retained hybrid folding after interacting with the Pt-tripod, as illustrated by a similar CD pattern (Fig. 1d and Supplementary Fig. 5). Moreover, the fluorescence spectra (Supplementary Fig. 6) showed that binding to Tel26 generated a blue shift accompanied by a four-fold intensity enhancement for the Pt-tripod.

The $^1H$ NMR titration of Pt-tripod with hybrid-2 human telomeric G-quadruplex (wtTel26) showed that Pt-tripod forms a 1:1 complex, but the spectral quality was not as good as that of hybrid-1 Tel26 G-quadruplex; however, no well-defined complex was formed with hybrid-2 wtTel26 at higher Pt-tripod equivalents (Supplementary Fig. 1a). The $^1H$ NMR titration of Pt-tripod against other G-quadruplexes (Supplementary Fig. 1b and 2–4) showed broadened and poorly resolved spectra, indicating less-defined binding of the Pt-tripod to these G-quadruplexes.

We have also evaluated the human telomerase inhibitory ability of the Pt-tripod using a telomeric repeat amplification protocol (TRAP-LIG) assay (Fig. 1e)[41]. The results showed that as the concentration of Pt-tripod increased, telomerase elongation products measured by PCR amplification clearly decreased, illustrating that the Pt-tripod is an effective telomerase inhibitor. The $IC_{50}$ value is determined to be $1.22 \pm 0.10\ \mu M$, which is lower than most of the G-quadruplex-targeting ligands that used the same TRAP-LIG method as previously reported[41–43].

**Formation of monomeric and higher-order complex structures.** To further investigate the interaction between Pt-tripod and Tel26 G-quadruplex, we carried out ESI-MS, native PAGE and FRET melting experiments. All of these experiments showed Pt-tripod gradually binds Tel26 and induces monomeric, dimeric, and multimeric Pt-tripod–Tel26 complex structures depending on the Pt-tripod/Tel26 ratio.

The ESI-MS was performed to study the binding stoichiometry information of Pt-tripod–Tel26 complexes. The ESI-MS exhibited clear ion peaks for multiple Pt-tripod–Tel26 complex structures at different Pt-tripod/Tel26 ratios (Fig. 2a–d and Supplementary Fig. 7–10), agreeing with the 1D NMR titration and the following native PAGE results. At a Pt-tripod/Tel26 ratio of 1.0, the dominant peaks of the 1:1 complex were observed (Fig. 2b and Supplementary Fig. 8). After adding 2.0 equivalents of Pt-tripod, the major peaks for the 1:1 and 2:1 Pt-tripod–Tel26 complex were detected (Fig. 2c and Supplementary Fig. 9). Upon further titration of 3.0 Pt-tripod equivalents, we observed clear ion peaks for the dimeric 3:2 and 4:2 Pt-tripod–Tel26 complexes, in addition to ion peaks from the 1:1 and 2:1 complexes (Fig. 2d and Supplementary Fig. 10). This is consistent with the native PAGE result (Fig. 2e, f), which demonstrated small quantities

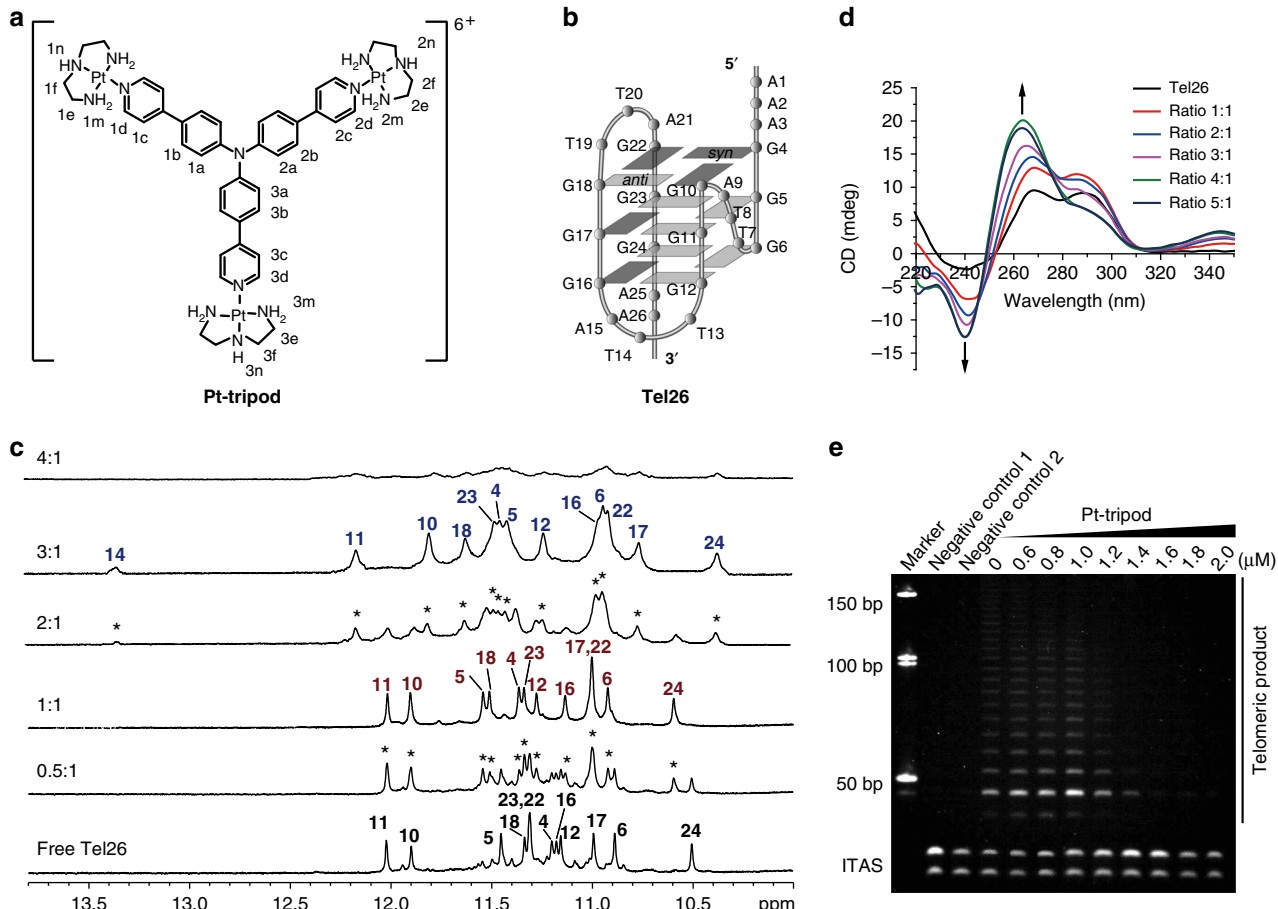

**Fig. 1** The Pt-tripod binds the intramolecular hybrid-1 human telomeric G-quadruplex and inhibits telomerase activity. **a** The chemical structure of the Pt-tripod with proton numbering. **b** The folding topology of the hybrid-1 human telomeric G-quadruplex adopted by Tel26 in K+ solution[44]. Deep gray box = (syn) guanine, light gray box = (anti) guanine. **c** Imino proton regions of the 1D $^1$H NMR titration spectra of Tel26 interacting with Pt-tripod in 100 mM K+, pH 7.0 solution, at 25 °C. Ratios of Pt-tripod/Tel26 are shown in the spectra. The imino assignments are labeled. Peaks arising from the new complexes are marked with asterisks. **d** CD spectra of the Tel26 titrated by the Pt-tripod in 100 mM K+ solution. Ratios of Pt-tripod/Tel26 are shown in the spectra. **e** The Pt-tripod showed excellent telomerase inhibition property by the TRAP-LIG assay. The band labeled as ITAS is an internal control primer. The $IC_{50}$ value is determined to be 1.22 ± 0.10 μM. Error value represents the standard deviation, $n = 3$

monomeric species at a Pt-tripod/Tel26 ratio of 3.0. Moreover, higher-order assembly structures seem to be unstable in the gas phase and will break into smaller fragments.

Next, native PAGE was used to explore the formation of higher-order structures depending on the Pt-tripod ratios (Fig. 2e, f). The gel showed that free Tel26 is a monomeric structure and the band appears red due to ethidium bromide staining under UV light. Upon titration of the Pt-tripod into Tel26, a slightly down-shifted band appeared at a Pt-tripod/Tel26 ratio of 1.0, indicating the formation of a monomeric 1:1 Pt-tripod–Tel26 complex. This band appears yellow due to the fluorescence of the Pt-tripod. At 2.0 Pt-tripod equivalents, a band of green fluorescence with slower mobility was observed at ~35 bp, corresponding to a dimeric structure. At 3.0 equivalents of Pt-tripod, this green band became deeper and dominant. Further titration to Pt-tripod/Tel26 ratios of 4.0 and 5.0 generated green bands for multimeric structures with even slower mobility at high molecular weight of ~50 bp and 100 bp. Consistent with the $^1$H NMR titration data, wtTel26 G-quadruplex forms a 1:1 complex with Pt-tripod at 1.0 equivalent but higher-order structures at higher Pt-tripod equivalents, whereas higher-order assemblies were observed for bcl2Mid and MycG4 G-quadruplexes at all Pt-tripod equivalents in the native PAGE (Supplementary Fig. 11).

We then used FRET assay to study the stabilizing ability of the Pt-tripod on various G-quadruplexes and double-stranded DNA (Supplementary Fig. 12). For the hybrid-1 human telomeric G-quadruplex Tel26 (Supplementary Fig. 12a and Supplementary Table 2), at 1.0 equivalent of Pt-tripod, one clear melting temperature ($T_m$) of 54.5 °C was observed, which is 8.4 °C ($\Delta T_m$) higher than that of free Tel26, corresponding to the monomeric 1:1 complex (Fig. 2e, f). At 2.0 and 3.0 Pt-tripod equivalents, a second $T_m$ was observed at around 87.6 °C, 41.5 °C ($\Delta T_m$) higher than that of the free Tel26, corresponding to the dimeric complex observed in the native PAGE, in addition to the 1:1 complex ($T_m$ of 54.5 °C) which was still present (Fig. 2e, f). At Pt-tripod equivalents at 4.0 and 5.0, a $T_m$ of >90 °C was observed, correlating to the higher-order multimeric structures observed in the native PAGE (Fig. 2e, f). For the hybrid-2 wtTel26 telomeric G-quadruplex (Supplementary Fig. 12b and Supplementary Table 2), consistent with the native PAGE results (Supplementary Fig. 11a), a 1:1 Pt-tripod complex was observed with the melting temperature of 54.0 °C, while the higher-order multimeric structures showed a $T_m$ of 86.8 °C. The bcl2Mid G-quadruplex doesn't appear to form a specific 1:1 complex even at the 1.0 equivalent of Pt-tripod, but a mixture of the free DNA and higher-order structures, which exhibited a $T_m$ of 85.0 °C

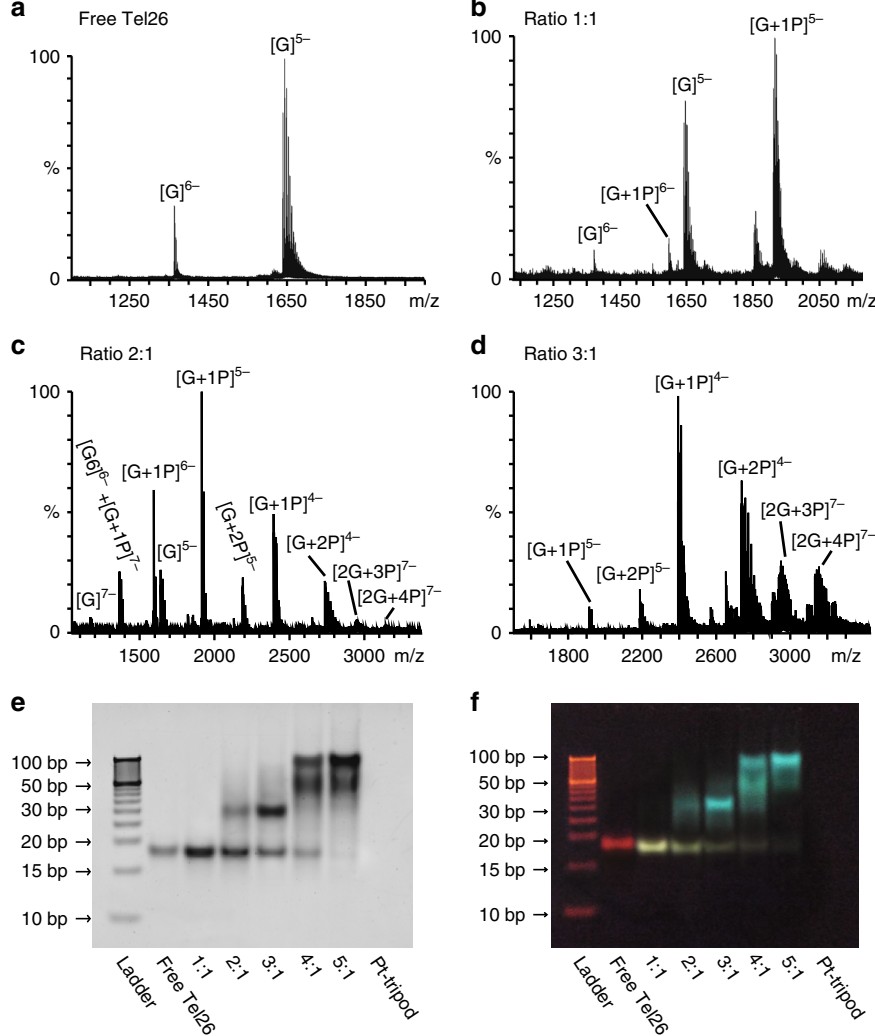

**Fig. 2** The formation of monomeric and higher-order Pt-tripod–Tel26 complex structures. **a–d** The ESI-MS of the Pt-tripod interacting with the Tel26 with peaks labeled. 'G' means 'Tel26 Guadruplex', and 'P' means 'Pt-tripod'. **e, f** The native PAGE of different ratios of the Pt-tripod with the Tel26. The gel pictures were photographed by the FluorChem M imager (ProteinSimple) and visualized by the UV light, respectively. The ratios of Pt-tripod/Tel26 are labeled

(Supplementary Fig. 12c and Supplementary Table 2), consistent with the native PAGE results (Supplementary Fig. 11b). MycG4-showed robust higher-order structure formation starting from Pt-tripod equivalent as low as 0.5 (Supplementary Fig. 11c and 12d, and Supplementary Table 2).

In contrast, the binding of Pt-tripod to the double-stranded DNA only induced a small $\Delta T_m$ of 2.5 °C (Supplementary Fig. 12e and Supplementary Table 2). The FRET competition assay (Supplementary Fig. 13) showed that the stability of 1:1 Pt-tripod–Tel26 complex remains 81 and 51% after adding 10 and 100 folds (base pair) of calf thymus (CT) DNA, respectively, illustrating the Pt-tripod has higher binding affinity with human telomeric G-quadruplex DNA over double-stranded DNA. Therefore, consistent with the NMR data, Pt-tripod binds to hybrid-1 human telomeric G-quadruplex to form specific 1:1 and 4:2 complexes (see following NMR results), however, it interactions with other G-quadruplexes are less-defined and more prone to inducing higher-order structures.

**Solution structure of the 1:1 Pt-tripod–Tel26 complex.** To investigate how the Pt-tripod binds to the Tel26 in the

monomeric and even higher-order complex structures, and where the Pt-tripod binds in these complexes, as well as how the Tel26 assembles into higher-order structures, we performed 2D NMR structural studies.

Unambiguous proton assignment of bound Tel26 of the 1:1 Pt-tripod–Tel26 complex was obtained by assigning H1s and H8s and the sequential connectivity of H8-sugar protons (Fig. 3a, b, Supplementary Fig. 14–15 and Supplementary Table 3, 4)[44]. Proton resonances for the bound Pt-tripod in the 1:1 complex were unambiguously assigned using TOCSY, COSY and NOESY experiments (Supplementary Fig. 16–17 and Supplementary Table 3–4). A1, G4, G10, G16, G17, G22, and A25 adopt a *syn* glycosidic conformation, which is reflected by the strong intensities of the H8-H1′ NOE cross-peaks (Supplementary Fig. 15c). Based on the complete assignment, the residues from the 5′ G-tetrad (G4, G10, G18, and G22) exhibited the largest proton chemical shift changes (Fig. 3c), indicating Pt-tripod stacking at the 5′ G-tetrad. Notably, large proton chemical shift changes were also observed for 5′ loop and flanking regions (A1, A2, A3, T8, A9, T20, and A21), which revealed important conformational rearrangements and ligand interactions at the Pt-

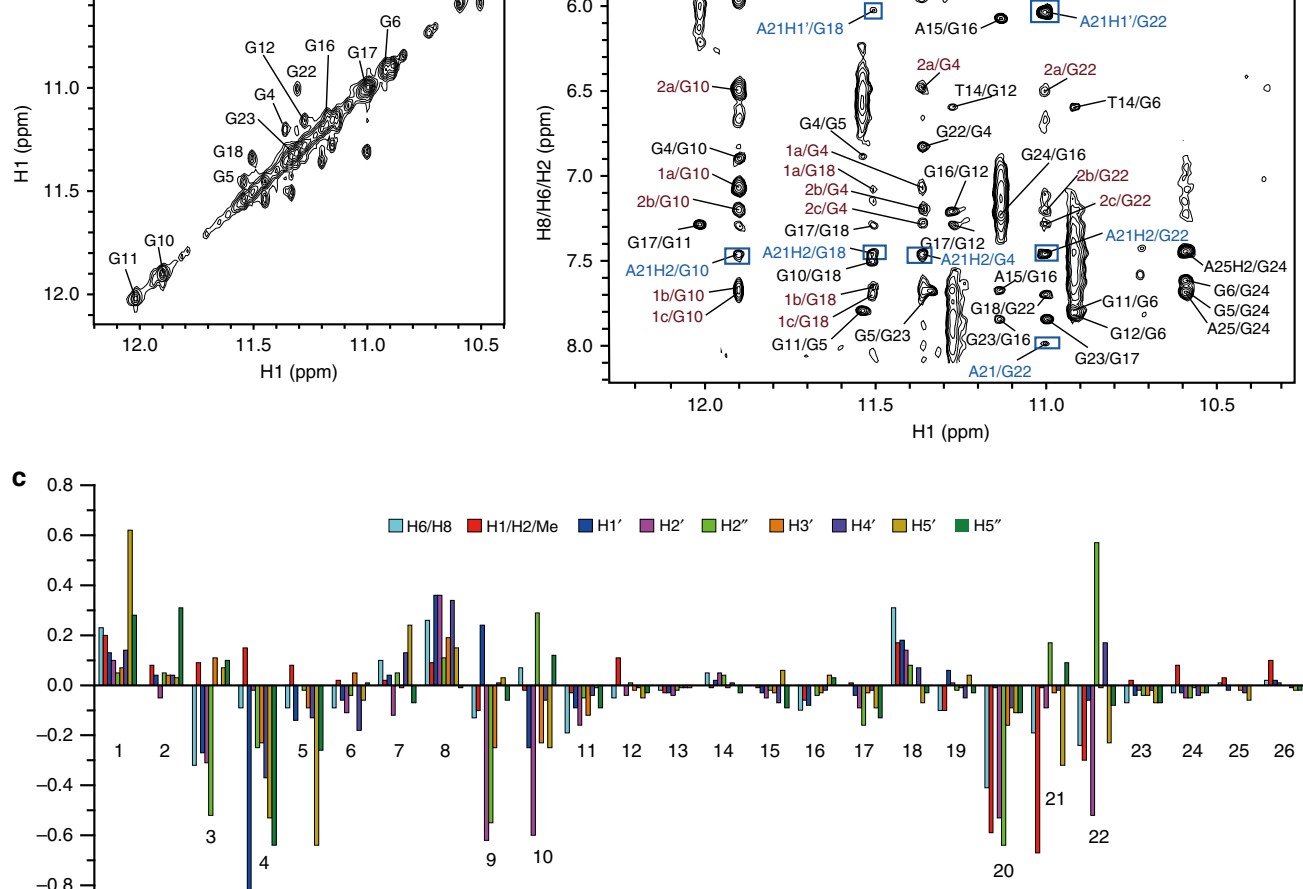

**Fig. 3** NMR spectra of the 1:1 Pt-tripod–Tel26 complex. **a** The expanded H1-H1 region of the NOESY at the 0.5:1 Pt-tripod/Tel26 ratio. Exchange cross-peaks between the free and bound Tel26 are labeled with residue numbers. **b** The expanded H1-H8/H6/H2 region of the 1:1 Pt-tripod–Tel26 complex. The $G_xH1$-$G_yH8$ cross-peaks of G-tetrads are colored in black. Inter-residue cross-peaks of A21 are framed and colored in blue. Intermolecular cross-peaks of the Pt-tripod and G-tetrad are colored in red. **c** The chemical shift difference of the protons between the free and bound Tel26 in the 1:1 complex. Condition: 100 mM $K^+$, pH 7.0, 25 °C

tripod binding site at the 5′-end of Tel26. We calculated the solution structure of the 1:1 Pt-tripod–Tel26 complex using NMR restraints[31], including 109 intermolecular NOEs between the Pt-tripod and Tel26 (Table 1 and Supplementary Table 5–7). The 15 lowest-energy refined structures (Fig. 4a and Supplementary Fig. 18) were selected and presented.

As displayed in the structure, due to Pt-tripod binding, the Tel26 undergoes a large conformational change at the 5′-end to form a well-defined binding pocket with Pt-tripod stacking on top of the 5′ G-tetrad (Fig. 4b). Binding of the Pt-tripod disrupts the A3-A9-A21 triple adenine capping structure at the 5′-end of the free Tel26. A21 from the last lateral loop was recruited by the Pt-tripod to form a triad stacking moiety with the first and second Pt-tripod pendant arms to cover the 5′-end (Fig. 4c). While A21 positioned above G18, the middle aromatic rings of the first and second Pt-tripod arms stack on top of G4 and G10 (Fig. 4c). A3 and A9 are shifted upwards, stacking above the second and third Pt-tripod arms respectively, and together with T20 form an A3-A9-T20 capping triad covering the newly formed A21–Pt-tripod plane nicely (Fig. 4d, e). This well-defined Pt-tripod binding pocket is supported by the inter-residue and intermolecular

NOEs observed between the protons of 5′-end residues and Pt-tripod (Fig. 3b and Supplementary Table 5–7).

The cationic platinum units of the first and second pendant arms stretch into the grooves (Fig. 4c) and are not as well defined, as shown by the more broadened peaks in 2D NMR spectra (Supplementary Fig. 17c). The first platinum unit of these two arms directs to the loop residue T19 for a potential hydrogen-bond interaction between the NH of Pt-tripod and the O2 of the T19 base (Fig. 4f). The second platinum unit is close to the negative phosphate backbones of G22 for electrostatic interactions between the $NH_2$ of Pt-tripod with the oxygen of backbone (Fig. 4g). The third pendant arm does not stack with the 5′-tetrad and almost entirely stretches to the groove between G4 and G10, important for the stabilization of chain-reversal loop residues T8 and A9 (Fig. 4d). The A9 stacks onto the first aromatic ring of the third pendant arm, while the T8 residue moved close to the platinum unit of the third pendant arm with a potential hydrogen bond between the O4 of the T8 base and the NH of the platinum unit (Fig. 4h). These interactions of the Pt-tripod and Tel26 are supported by the observation of intermolecular NOEs between Pt-tripod with loop residues (Supplementary Table 6).

**Table 1 Structural statistics for the solution structures of the 1:1 and 4:2 Pt-tripod–Tel26 complexes**

|  | 1:1 Complex | 4:2 Complex |
|---|---|---|
| NMR distance and dihedral constraints |  |  |
| Distance restraints |  |  |
| Total NOE | 652 | 771 |
| Intra-residue | 287 | 256 |
| Inter-residue | 208 | 262 |
| Sequential ($|i - j| = 1$) | 148 | 192 |
| Nonsequential ($|i - j| > 1$) | 60 | 70 |
| Hydrogen bonds | 24 | 30 |
| Pt-tripod (Intramolecular) | 24 | 24 |
| Intermolecular (Pt-tripod/Tel26) | 109 | 199 |
| Total dihedral angle restraints | 14 | 18 |
| Structure statistics |  |  |
| Violations (mean and s.d.) |  |  |
| Distance constraints (Å) | 0.033 ± 0.022 | 0.047 ± 0.037 |
| Dihedral angle constraints (°) | 0.765 ± 0.113 | 0.994 ± 0.324 |
| Max. dihedral angle violation (°) | 10.39 | 10.11 |
| Max. distance constraint violation (Å) | 0.430 | 0.564 |
| Deviations from idealized geometry |  |  |
| Bond lengths (Å) | 0.013 ± 0.012 | 0.019 ± 0.014 |
| Bond angles (°) | 1.530 ± 0.040 | 1.453 ± 0.025 |
| Impropers (°) | 0.985 ± 0.024 | 0.939 ± 0.016 |
| Average pairwise r.m.s. deviation[a] (Å) |  |  |
| G-tetrad | 0.55 ± 0.17 | 0.75 ± 0.26 |
| All | 1.33 ± 0.38 | 1.65 ± 0.36 |

[a]Pairwise r.m.s. deviation was calculated among 15 refined structures

**Solution structure of the 4:2 Pt-tripod–Tel26 complex**. As the [1]H NMR peaks at 3.0 Pt-tripod equivalents were well resolved and suitable for NMR structural analysis (Fig. 1c), we solved the structure to understand how the Pt-tripod binds the Tel26 at high ligand ratios. Surprisingly, the major complex structure at the 3:1 ratio is a unique 4:2 Pt-tripod–Tel26 dimeric structure, in accordance with other experimental data including ESI-MS and native PAGE.

The 1D [1]H NMR spectrum of the dimeric structure has 14 peaks at 5 and 15 °C, with two thymine imino peaks observed, indicating hydrogen-bonded conformation (Supplementary Fig. 19a, c). We unambiguously assigned these two imino peaks using [1]H-[15]N HMQC experiments with low-enrichment (6%) site-specific [15]N-labeled thymines at 5 °C (Supplementary Fig. 19b). Complete assignment of the 4:2 dimeric complex was achieved using NOESY, TOCSY, and COSY (Fig. 5a, b, Supplementary Fig. 19–22 and Supplementary Table 8–9). As only one set of proton resonances was observed for the 4:2 complex, a twofold symmetry is indicated in this dimeric G-quadruplex molecular system. For each G-quadruplex subunit, similar to the 1:1 complex, strong H8-H1′ NOE cross-peaks were observed for A1, G4, G10, G16, G17, and G22, indicating a *syn* glycosidic conformation (Supplementary Fig. 21b). Moreover, T14 in the 3′-loop was observed to transfer from the *anti* to the *syn* glycosidic conformation, an unexpected difference from the 1:1 complex (Supplementary Fig. 15c, 21b). A25 in the 3′-flanking inversely changed, from the *syn* to the *anti* glycosidic conformation (Supplementary Fig. 15c, 21b). Residues from the 3′ G-tetrad (G6, G12, G16, and G24) and 3′ loop and flanking regions (T13, T14, A15, A25 and A26) displayed the largest proton chemical shift changes (Fig. 5c), indicating the 3′-end is the second binding site for the Pt-tripod and that the binding of Pt-tripod causes a

large conformation reorientation in the 3′-capping regions. The binding of the second Pt-tripod is supported by the TOCSY and NOESY spectra, which showed two sets of crosspeaks from two bound Pt-tripods (Supplementary Fig. 22), revealing another Pt-tripod interacted with the 3′ of Tel26 G-quadruplex.

We calculated the solution structure of the unique 4:2 Pt-tripod–Tel26 dimeric complex using NMR restraints, including the intermolecular NOEs between Pt-tripod and Tel26 and between the two subunits of the dimeric complex (Table 1 and Supplementary Table 10–13). The ensemble of the 15 lowest-energy structures after distance-restrained refinement are superimposed and presented (Fig. 6a and Supplementary Fig. 23) with an RMSD of 1.65 Å. As presented, the dimeric architecture is interlocked symmetrically by the 3′–3′ interface (Fig. 6a, b). For each 2:1 Pt-tripod–Tel26 subunit, a second Pt-tripod binds to the 3′-end of Tel26, with the 5′-end of the complex retaining the same conformation as the 1:1 complex. Notably, the proton peaks of the 3′-Pt-tripod in the 4:2 complex are sharper than the 5′-Pt-tripod (Supplementary Fig. 22), suggesting a less dynamic conformation of the 3′-Pt-tripod and illustrating that the 3′-Pt-tripod is better bound in the 4:2 complex.

In the 4:2 complex structure, the Pt-tripod binding to the 3′-end is well defined (Fig. 6c, d), with a similar binding pattern as the 5′-end. As shown by the complex structure, the Pt-tripod interacts with the Tel26 not only through π–π stacking between the aromatic core and the 3′ external G-tetrads (Fig. 6c) but also through hydrogen bonding and electrostatic interactions of the cationic platinum units with DNA backbones (Fig. 6e, f). Similar to the 5′-end binding, A15 from the second lateral loop was recruited by Pt-tripod to form a triad stacking moiety with the first and second Pt-tripod arms to cover the 3′-external tetrad, with A15 stacking on top of G12 and G16, and the first and second Pt-tripod arms stacking on top of G6 and G24, respectively (Fig. 6c). The cationic platinum units of these two pendant arms stretch into the groove and potentially interact with the negative phosphate backbones of T13 and G16 of Tel26, respectively (Fig. 6e, f). Again, the third pendant arm stretched out entirely into the groove between G6 and G24 (Fig. 6c), however, unlike Pt-tripod bound at the 5′-end, no clear interactions with DNA bases were observed.

**Ligand-induced T13:A25:T14 hydrogen-bonded triad**. A significant conformational rearrangement distinct from that of the free Tel26 was observed at the 3′-end of Tel26 upon Pt-tripod binding. As expected, the T14:A25 capping structure at the 3′-end was destroyed and rearranged. The recruited A15 stacks over the center of G12 and G16, joining the Pt-tripod to form a plane covering the 3′ G-tetrad (Fig. 6c). Surprisingly, based on the strong inter-residue NOEs from T13H3/A25H2 and T14H3/A25H8 (Fig. 5b), we observed a stable T13:A25:T14 hydrogen-bonded triad formation (Fig. 6d). The hydrogen-bonded T:A:T triad stacked very well on the A15–Pt-tripod plane to stabilize the dimeric structure. Formation of the T13:A25:T14 triad by hydrogen bonding also explains the *anti* and *syn* glycosidic conformational conversion of T14 and A25 mentioned above.

**A26:A26* non-canonical pair interlocking dimeric structure**. Interestingly, the anchoring of two 2:1 Pt-tripod–Tel26 subunits to build the 3′–3′ dimeric structure was achieved by the formation of an A26:A26* non-canonical base pair with A26s from two different Tel26 subunits (Fig. 6g, h). The interlocking interface of the A26:A26* non-canonical pair at the 3′-end in the 4:2 complex structure constitutes an interesting feature of the dimeric architecture. This interlocked dimeric arrangement was supported by the discovery of inter-residue NOEs between the A26 and T13,

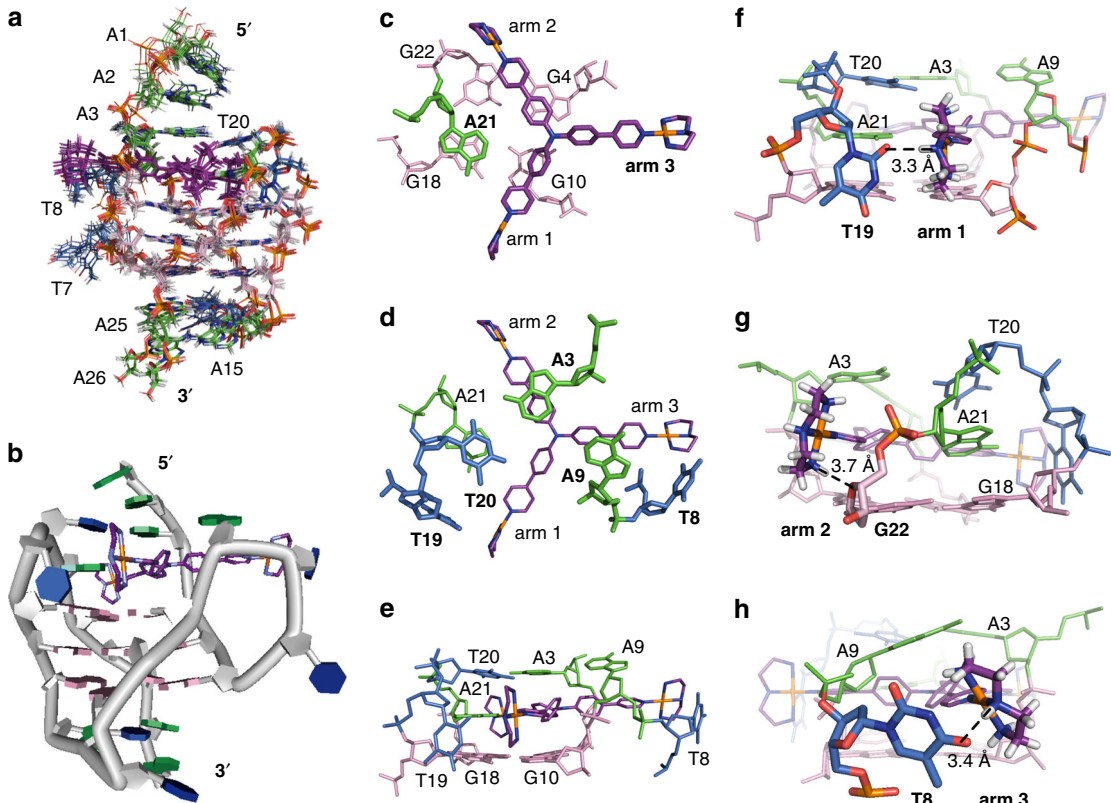

**Fig. 4** Solution structure of the 1:1 Pt-tripod–Tel26 complex (PDB code: 5Z80). **a** The ensemble of the superimposed 15 NMR restraint-refined structures of the 1:1 complex. **b** A representative structure of the 1:1 complex. **c–e** Three different views of the Pt-tripod-induced binding pockets at the 5′-end with residues labeled. **f–h** Potential hydrogen bonding and electrostatic interactions of three platinum units of Pt-tripod and Tel26 at the 5′-end, respectively. The distance between the NH of the first platinum unit and the O2 of the T19 base (N-O) is 3.3 Å (**f**). The distance between the NH₂ of the second platinum unit and the oxygen of the G22 ribodesose (N-O) is 3.7 Å (**g**). The distance between the NH of the third platinum unit and the O4 of the T8 base (N-O) is 3.4 Å (**h**). Pt-tripod molecules are colored in purple. Guanines are colored in pink. Adenines are colored in green. Thymines are colored in blue

T14 protons (Supplementary Fig. 24 and Supplementary Table 13). Moreover, clear NOE connectivity patterns from continuous DNA residues were observed for the 3′-flanking A25-A26 segment (Supplementary Fig. 21a, b), which revealed that A26 stacked above A25.

## Discussion

This work provides structural information regarding the dynamic binding of the Pt-tripod with the hybrid-1 human telomeric G-quadruplex Tel26 (Supplementary Movie 1). Structural analysis indicated the binding of Pt-tripod at the 5′-end is more favoured than the 3′-end. Upon Pt-tripod binding, the Pt-tripod recruits A21 to form the A21–Pt-tripod plane, stacking on top of the 5′-external G-tetrad and locking the position of Pt-tripod. The A21–Pt-tripod plane is further covered and stabilized by the ligand-induced A3-A9-T20 triad. Moreover, loop residues T8 and T19 are also rearranged to interact with two platinum units through hydrogen bonding, respectively. At higher equivalence of Pt-tripod, the second Pt-tripod molecule binds the 3′-end of Tel26 and induces a dimeric G-quadruplex complex structure interlocked by an atypical A:A base pair at the 3′–3′ interface. The A15 at the 3′ end is recruited by the second Pt-tripod to form an A15–Pt-tripod plane, which is further covered by a hydrogen-bonded T13:A25:T14 triad. This interlocked, multi-layer binding pocket of two symmetrical bound-ligands has not been seen in reported molecular structures of G-quadruplex–small molecule complexes. The

overlapped structures of the free Tel26, 1:1, and 4:2 complexes were displayed (Supplementary Fig. 25).

By solving the 1:1 and 4:2 Pt-tripod–Tel26 complex structures, we found the Pt-tripod possesses large structural area with three pendant arms for multiple interaction modes, including π–π stacking, hydrogen bonding and electrostatic interactions. We found the interactions of loop and flanking residues with the bound Pt-tripod play an important role in the formation and stabilization of these two complex structures. As shown in the 5′ complex, the Pt-tripod possesses a non-canonical structure with a non-planar tertiary amine conformation and remains non-planar in the complex structure. This is different from the reported typical cyclic fused[30,34] and crescent-shaped molecules[31] with a planar aromatic central core for strong π–π stacking with G-tetrads, which is normally considered to be the essential binding mode for G-quadruplex ligands to achieve high affinity to G-quaduplexes. These phenomena are very inspiring to the design of G-quadruplex-targeting compounds as potential anticancer drugs. The unique binding mode of Pt-tripod includes first utilizing its two arms to recruit an adenine to form a Pt-tripod-A plane covering the external G-tetrad, with the two cationic platinum units interacting with loop residues and negatively-charged phosphate backbones of two grooves, and then utilizing the third arm to further lock the Pt-tripod binding position by similar interactions with the third groove and stabilization of the chain-reversal loop.

To date, all reported solution structures of intramolecular G-quadruplex–small molecule complexes have been monomeric, i.e.,

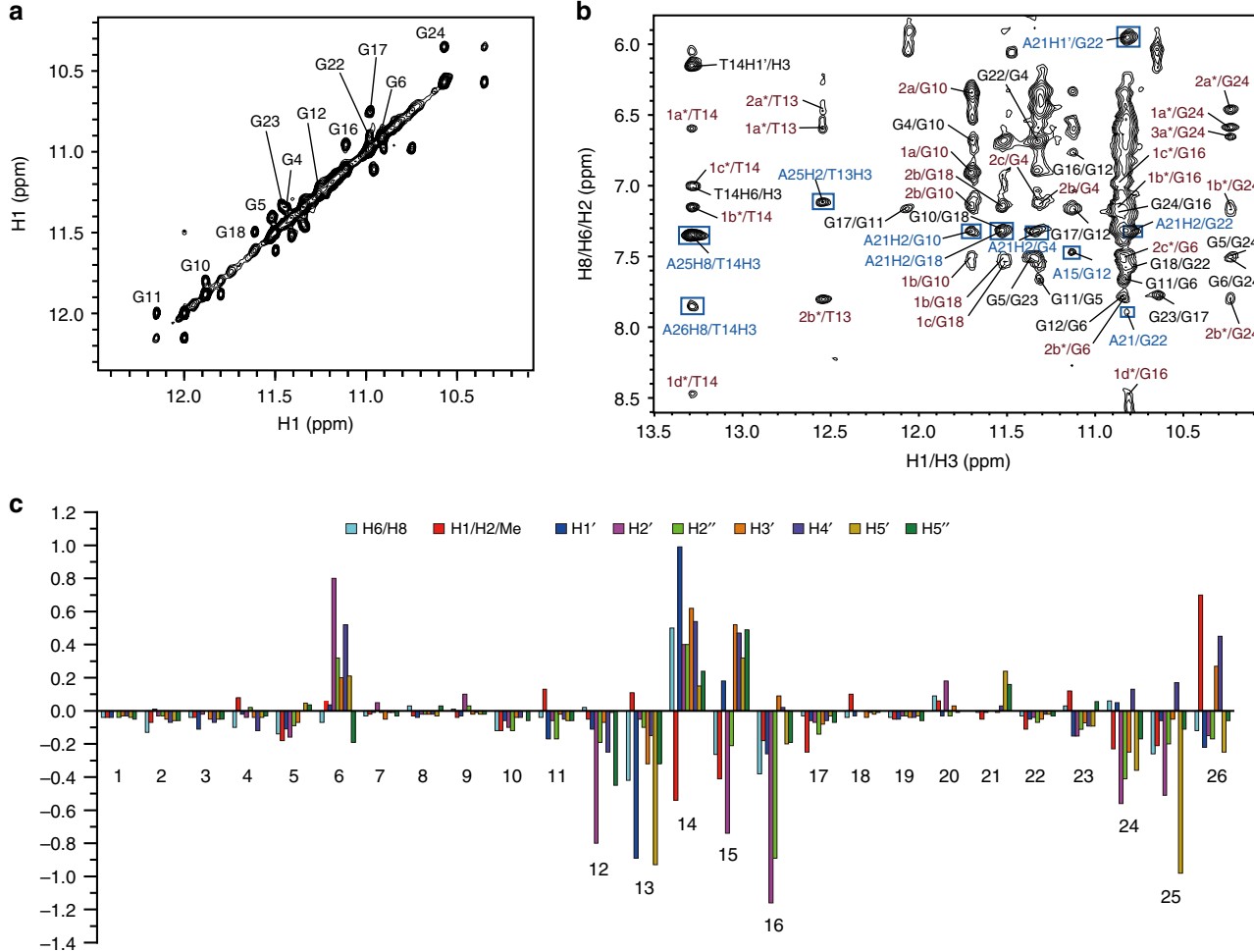

**Fig. 5** NMR spectra of the unique dimeric 4:2 Pt-tripod–Tel26 complex. **a** The expanded H1-H1 region at a Pt-tripod/Tel26 ratio of 2.0 at 25 °C. Exchange cross-peaks of Tel26 between 1:1 and 4:2 complexes are labeled. **b** The expanded H1/H3-H8/H6/H2 region of the 4:2 complex at 15 °C. The $G_xH8$-$G_yH1$ NOEs of G-tetrads are colored in black. Inter-residue NOEs of loop/flanking residues are framed and colored in blue. Intermolecular NOEs between the Pt-tripod and G-tetrad are in red. Assignments belonging to the 5′-Pt-tripod and 3′-Pt-tripod are labeled without or with asterisks. **c** The chemical shift difference of Tel26 protons between the 1:1 and 4:2 Pt-tripod–Tel26 complex at 25 °C. Condition: 100 mM K+, pH 7.0

one or two molecules bound to one intramolecular G-quadruplex[30–36]. Here, the Pt-tripod induces a well-defined dimeric 4:2 complex structure. Compared with the structural details of the 5′ and 3′ binding pockets, we found the 3′ binding pocket is better-defined than the 5′ binding pocket. The T13:A25: T14 capping triad at the 3′-end is more stable as it is connected with hydrogen bonding, while the A3-A9-T20 capping triad at the 5′-end is more flexible. The more stable hydrogen-bonded T13:A25:T14 triad may also facilitate stacking and assembling into the dimeric structure.

In summary, we found the non-planar Pt-tripod, a tertiary amine with cationic platinum groups, specifically binds the hybrid-1 telomeric G-quadruplex. The binding and assembling of Pt-tripod with Tel26 G-quadruplex is achieved by multiple arms with properly sized cationic platinum units and a combination of multiple interaction modes including the π–π stacking, hydrogen bonding, and electrostatic interactions. The Pt-tripod interacts and stabilizes the loop and flanking regions around binding pockets, which is important for the tight and specific binding and further assembling of higher-order G-quadruplex structures. The stable T:A:T triad with hydrogen bonding induced by the large Pt-tripod scaffold enables further stacking and base pairing of A:A pair to build the 3′-3′ dimeric complex structure. This study thus provides a structural

basis for designing non-planar compounds as anticancer drug candidates targeting human telomeric G-quadruplexes and offers breakthrough knowledge on the interactions between Pt complexes and various G-quadruplex topologies.

## Methods

**Sample preparation**. The Pt-tripod was synthesized as described in the Supplementary Methods. DNA oligonucleotides were purchased from Sangon Biotech (Shanghai) Co., Ltd with PAGE or HPLC purification. The low-enrichment (6%) site-specific 15N-labeled oligonucleotides were synthesized using the 15N-labeled dG-phosphoramidite purchased from Cambridge Isotope Laboratories (USA). DNA G-quadruplex structures were forming through annealing of DNA oligonucleotides in buffer. DNA samples were annealed by heating at 95 °C for 5 min and then slowly cooling to room temperature overnight. DNA concentration was measured by Nanodrop 200/200c (Thermo Science).

**NMR experiments**. All NMR experiments were performed on the Bruker AVIII 600 MHz and AVIII 700 MHz (equipped with a cryprobe) spectrometers. 25 mM K-phosphate and 70 mM KCl buffer (pH 7.0) in D₂O/H₂O (10%/90%) or D₂O (99.8%) were used to prepare DNA samples. The final concentrations of DNA samples were 0.1–2.5 mM. The 1H-15N HMQC experiments were used to assign the hydrogen bonding thymine imino protons. 2D NMR experiments, including COSY, TOCSY and NOESY, were collected at different temperatures of 5, 15, and 25 °C in H₂O and D₂O solution for the 1:1 and 4:2 Pt-tripod–Tel26 complexes. The mixing times of NOESY were set from 50–300 ms, and TOCSY at 40 ms. WATERGATE or

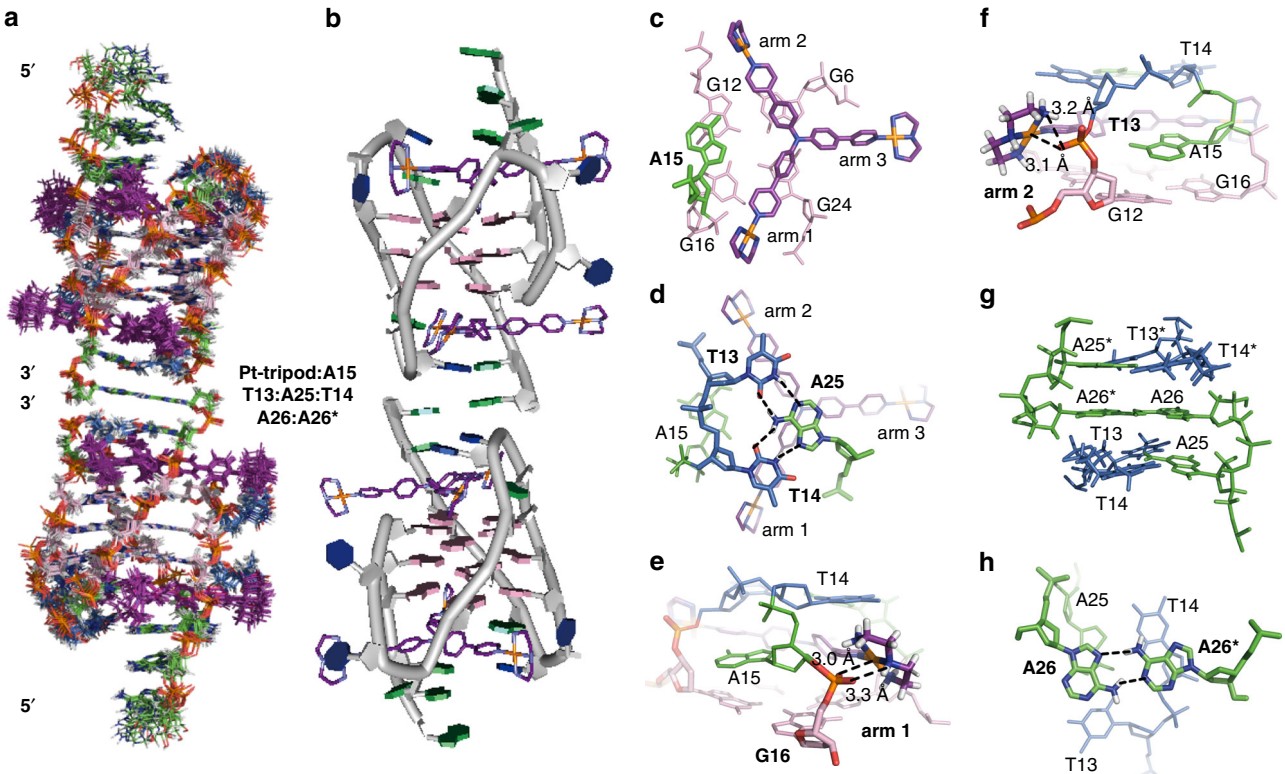

**Fig. 6** Solution structure of the unique dimeric 4:2 Pt-tripod–Tel26 complex (PDB code: 5Z8F). **a** The ensemble of the superimposed 15 NMR restraints-refined structures of the dimeric 4:2 complex. **b** A representative structure of the dimeric 4:2 complex. **c, d** Two different views of the Pt-tripod-induced binding pockets at the 3′-end with residues labeled. **e, f** Electrostatic interactions between the platinum units of the first and second Pt-tripod arms with the negative backbones of Tel26 at the 3′-end. The distance between the NH of the first platinum unit and the oxygen of the G16 backbone (N-O) is 3.3 Å; the distance between the Pt of the first platinum unit and the oxygen of the G16 backbone (Pt-O) is 3.0 Å (**e**). The distance between the NH₂ of the second platinum unit and the oxygen of the T13 backbone (N-O) is 3.2 Å; the distance between the Pt of the second platinum unit and the oxygen of the T13 backbone (Pt-O) is 3.1 Å (**f**). **g, h** Two different views of the 3′–3′ interface of the dimeric complex structure with residues labeled. Hydrogen bonding between T13:A25, T14:A25, and A26:A26* are labeled. The Pt-tripod molecules are colored in purple. Guanines are colored in pink. Adenines are colored in green. Thymines are colored in blue. Residues from the other subunits are labeled with asterisks

presaturation water suppression techniques were used for water NMR samples. Peak assignments and integrations were made using Sparky (UCSF).

**NOE-distance restrained molecular dynamics simulation**. Proton Distances were obtained based on the NOE cross-peaks integrated at NOESY spectra (50–300 ms mixing times) with ±20% variance. The peak volumes were referenced by the distance of Me-H6 (2.99 Å) in thymine. Structure calculations were performed in the program X-PLOR[45] and Accelrys Discovery Studio 2.5.5[46]. The Pt-tripod molecule was geometry optimized and calculated the partial atomic charges using the Gaussian 03[47]. The topology and parameters of the Pt-tripod were generated by hand. The starting models of the 1:1 and 4:2 Pt-tripod–Tel26 complex were constructed in Discovery Studio, with the conformations deduced from the NOE data. These starting models were first minimized and equilibrated, and then soaked into water solvent using 15 Å water layer to run dynamic simulation in Accelrys Discovery Studio. CHARMM force field was used for the calculations. NOE-restrained simulation annealing refinement calculations were performed in XPLOR. Dihedral angle restraints of 60 (±40)° and 240 (±40)° with a force constant of 10 kcal mol⁻¹ rad⁻² were restrained to *syn* and *anti* nucleotides, respectively. NOE and hydrogen bond force constants were gradually scaled to 10 and 20 kcal mol⁻¹ Å⁻², respectively. The complete molecular systems were subjected to 1000 steps of energy minimization. The structures were then equilibrated at 1000 K for 10 ps. Subsequent restrained molecular dynamics and cooling simulation were carried out with temperature reduced by 25 K with 1000 time steps of 2 fs each cycle until the final temperature reached 300 K. The structures were further subjected to 1000 steps of energy minimization. The 15 best structures were chose according to the minimum energy and number of NOE violations.

**Mass spectrometry**. Negative electrospray ionization mass spectrometry (ESI-MS) was performed on the Xevo G2 QTOF (Waters) or solariX ESI (Bruker) mass spectrometer. G-quadruplex DNA samples were in 10 μM concentration with 100

mM ammonium acetate buffer (pH 7.0). Different equivalents of the Pt-tripod were added and incubated for 2 h before measurement.

**Native polyacrylamide gel electrophoresis (PAGE)**. Native PAGE experiments were carried out with a native gel containing 18% acrylamide in 1X TBE buffer (pH = 8.0). G-quadruplex DNA samples were prepared in 25 mM K-phosphate, 70 mM KCl buffer (pH 7.0) with different ratios of the Pt-tripod. Each sample contains 0.3 nmol DNA. DNA bands were stained by ethidium bromide. The gels were first visualized by UV light recorded by a digital camera, and then photographed in the FluorChem M imager (ProteinSimple).

**Telomeric repeat amplification protocol (TRAP-LIG) assay**. dNTP mix, RNase inhibitor, and Taq polymerase were purchased from TaKaRa Biotechnology. DNA oligonucleotide primers were synthesized from Sangon Biotech (Shanghai Co., Ltd.Firstly, telomerase extract was prepared from HeLa cells using the NP-40 lysis buffer (10 mM Tris-HCl, pH 8.0, 1.0 mM MgCl₂, 1.0 mM ethylene glycol tetraacetic acid (EGTA), 1.0% NP-40, 0.25 mM sodium deoxycholate, 10% glycerol, 150 mM NaCl, 0.1 mM phenylmethanesulfonylfluoride (PMSF) and 5.0 mM β-mercaptoethanol (β-ME)). Each reaction was performed in a final volume of 50 μL consisting of 10× TRAP buffer (5 μL), bovine serum albumin (BSA, 0.05 μg per sample, 1.0 μL), dNTP mix (4.0 μL, 2.5 mM), TS primer (1.0 μL of 100 ng μL⁻¹), RNase inhibitor (0.5 μL, 2 U μL⁻¹) and different concentrations of Pt-tripod (5 μL). Then the mixtures were transferred to a thermal cycler (Bio-Rad S1000, USA) for the initial elongation step: 30 °C (30 min), followed by 94 °C (10 min) and a final maintain at 20 °C. Secondly, the QIA quick nucleotide purification kit (Qiagen, 28304) was used to purify the elongated product and remove the extra Pt-tripod. Finally, the purified extended samples were subjected to PCR amplification again: 94 °C for 90 s, followed by 35 cycles of PCR reaction: 95 °C for 30 s, 50 °C for 30 s and 72 °C for 60 s. PCR master mix consisted of 10× TRAP buffer (5.0 μL), BSA (1.0 μL, μg sample⁻¹), dNTP mix (4.0 μL, 2.5 mM), TS primer (1.0 μL, 100 ng μL⁻¹), primer mix (1.0 μL; ACX reverse primer 100 ng μL⁻¹, NT primer 100 ng μL⁻¹, and

TSNT internal control primer $4.0 \times 10^{-14}$ M), Taq polymerase (0.4 μL, 5 U μL$^{-1}$), and purified extended samples (37.6 μL). DNA products were resolved on 8% polyacrylamide gel and visualize under UV illumination.

**CD spectroscopy**. CD spectra were measured using a J-810 spectropolarimeter (JASCO, Japan) with a wavelength range of 220–360 nm, 1 cm optical path-length, and 200 nm min$^{-1}$ scan speed at room temperature. G-quadruplex DNA samples were dissolved in 10 mm Tris-HCl (pH 7.4) in the presence of 10 mM KCl or in the absence of metal ions. The concentration of DNA samples was 3 μM. After each addition of the Pt-tripod, the solution was stirred and allowed to equilibrate for at least 5 min. Data analysis was performed using Origin 8.5 (OriginLab Corp.).

**Fluorescence resonance energy transfer (FRET) assay**. The fluorescently labeled oligonucleotides were dissolved in deionized water to make a 40 μM stock solution. Then the stock solution was diluted in 60 mM potassium cacodylate buffer (pH 7.4). G-quadruplex structures were forming through annealing of DNA oligonucleotides in buffer. Fluorescence melting curves were recorded with a Roche Light Cycler 2.0 real-time PCR machine, using a total reaction solution of 20 μL, which contains different concentration of Pt-tripod and 400 nM labeled oligonucleotides, as well as different concentration of CT DNA (just for competition assay). Measurements were carried out with excitation at 470 nm and detection at 530 nm. Fluorescence readings were recorded at intervals of 1 °C over the range of 37–99 °C, with a constant temperature being maintained for 30 s ahead of each recording to ensure a stable value. Final data analysis was carried out using Origin 8.5 (OriginLab Corp.).

The oligonucleotides for FRET melting measurements were fluorescently labeled with FAM at the 5′-end and TAMRA at the 3′-end (FAM: 6-carboxyfluorescein, TAMRA: 6-carboxytetramethylrhodamine). The concentration of CT DNA was measured by Nanodrop 200/200c (Thermo Science) using extinction coefficient and was reported per base pair[48].

**Fluorescence studies**. Fluorescence experiments were performed on an Edinburgh FLS 920 spectrometer (UK) at room temperature. G-quadruplex DNA samples were prepared in 10 mM Tris-HCl, 100 mM KCl buffer (pH 7.4). DNA stock solutions were successively added into the Pt-tripod solution (1 μM) in 10 mM Tris-HCl buffer (pH 7.4) with 100 mM KCl. The complex solution was stirred and allowed to equilibrate for at least 5 min. The emission spectra were measured using a 10 mm path length quartz cuvette with an excitation wavelength of 405 nm and recorded at a range of 420–800 nm.

**Data availability**. The coordinates for structures of the 1:1 Pt-tripod–Tel26 complex (accession codes: 5Z80) and 4:2 Pt-tripod–Tel26 complex (accession codes: 5Z8F) have been deposited in the Protein Data Bank. All data that support the findings of this study are available from the corresponding author upon reasonable request.

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

## Acknowledgements

This work was funded by the 973 program (Nos. 2014CB845604 and 2015CB856301), the National Science Foundation of China (Nos. 21837006 and 21572282), the Ministry of Education of China (IRT-17R111), Science and Technology Planning Project of Guangdong Province (Nos. 2013B051000047 and 207999), the Fundamental Research Funds for the Central Universities, and the US National Institutes of Health (R01CA177585 (D.Y.), and P30CA023168 (Purdue Center for Cancer Research)). We thank Professor Hai-Bin Luo (School of Pharmaceutical Sciences, Sun Yat-Sen University) for his support with the molecular dynamics simulation. We thank Qian Hu (School of Life Science, Sun Yat-Sen University) for his help with the TRAP-LIG experiment.

## Author contributions

Z.W.M. and D.Y. conceived and directed the project. W.L. performed the NMR, CD, and native PAGE experiments, assigned the NMR spectra, and did structure refinements. Y.F.Z. synthesized the Pt-tripod compound. Y.F.Z. and W.L. performed the TRAP-LIG assay. C.T.S. and L.Y.L. performed the fluorescence and FRET experiments. Y.F.Z., W.Z., and F.W. performed the ESI-MS. W.L. wrote the paper with the help of Z.W.M. and D.Y.

## Additional information

**Competing interests:** The authors declare no competing interests.

