## [Peer Review File · Nature Communications]

Reviewers' comments:

Reviewer #1 (Remarks to the Author):

The manuscript by Yang, Mao and their colleagues is focused on structural studies of DNA G-quadruplex interactions with a platinum(II) complex (Pt-tripod). A specific G-quadruplex studied originates from the biologically relevant hybrid-1 human telomeric repeat (Tel26) which has been implicated in inhibition of telomerase activity. Most excitingly, authors uncovered two modes of interactions and solved the 1:1 and the first dimeric 4:2 Pt-tripod-Tel26 complex structures by NMR. Structure calculations are supported with a large number of intermolecular NOEs between the Pt-tripod and Tel26. Scientific quality of this manuscript is high. Presentation and discussion of results are clear and supported with experimental data. Findings described in this manuscript are original and will be of interest to a wide range of scientific communities. As a comment, it could be suggested that authors try to discuss importance of new structures in view of promising properties of Pt-tripods as drugs. How is equilibrium between 1:1 and 4:2 complexes controlled?

It is hard to follow the chemical shift differences in Figures 3c and 5c.

Reviewer #2 (Remarks to the Author):

In this manuscript, authors studied a trinuclear Pt complex with a non-planar tripod structure and the interactions between the complex and several wild type and unnatural G-quadruplex (GQ) folding sequences with biological significance. The binding modes and the structural conformation of Pt-GQ complexes were studied under various stoichiometry of Pt/GQ. With the most stable binding to a revised telomeric GQ sequence, Tel26, the author identified two Pt-GQ with monomeric and dimeric Tel26 GQ. The novel structure of a 4:2 Pt-Tel complex was characterized by 1D and 2D NMR. This high order binding structure of Tel26 with Pt complex was also supported by ESI-MS spectra and native PAGE mobility analysis. This study offered breakthrough knowledge on the interactions between Pt complexes as anticancer drug candidates and various GQ topologies as drug targets on two novel levels. First, the proposed Pt complex possess a non-canonical structure for GQ binder. The complex has no extensive planar aromatic area for strong π - π stacking with G-quartets, which was normally considered to be the essential binding modes for all types of GQ binder to achieve high thermal and/or dynamic stability of GQs. Second, the manuscript offers, to my knowledge, the first NMR structure of high-order multimeric GQs with various binding stoichiometry of binder molecules. This structure would provide pivotal supports from structural biology to advance the studies in the biology and therapeutics of GQ structures and GQ targeting drug molecules. The experimental results and reference are sufficient and well presented to support author's arguments. I would like to recommend the publication of the manuscript after the following minor concerns were addressed by the authors.

1. Figure S6 indicated that wtTel26 and Tel26 both showed a much higher thermal melting temperatures under high stoichiometric ratio of Pt vs GQ. The authors attribute the results as an evidence of the existence of high-order structures. However, the melting curves showed clearly two transitions in UV absorption intensity. The authors need to discuss in more details about the melting process. What is the structural transition occurring at these two transitions, respectively? Are the two transition a two step melting of the high-order structure or melting processes of two individual types of folding structures, one low-order monomeric structure and one high-order dimeric GQ structure? The thorough discussion on thermal melting curve would offer more information of folding/unfolding dynamics of the high-order Pt-GQ complexes.

2. The manuscript described the NMR structures of 1:1 and 4:2 Pt-Tel26 GQ complexes. In

discussion part, authors also claims that the three Pt cationic cores can fit into the grooves and lock the complex to GQ monomer and dimers via electrostatic attraction. However, in result part, the structural information of Pt cores in either 1:1 or 4:2 Pt-Tel26 complexes were mentioned. The claims that tripod Pt complex offer the unique interaction modes via Pt cationic cores with GQ grooves in the discussion part seems quite out of sudden. More result explanation on this issue would significantly strengthen the manuscript.

3. As the author briefly mentioned in the discussion, "Compounds with a more flexible central core are unlikely to strongly bind to G-quadruplexes. However, in this work, the Pt-tripod is a tertiary amine with a totally nonplanar conformation. Nevertheless, it specifically recognizes the hybrid-1 human telomeric G-quadruplex." The unique stable high-order structures were achieved by Pt-tripod complexes without the classic π - π stacking. The phenomenon is very inspiring to the design of GQ-targeting compounds and may open the structure library significantly. However, the discussion on elaborating the novel interaction modes offered by Pt-tripod complexes was quite limited in the current manuscript. To enhance the impact of the manuscript, I suggest the authors provide more thorough discussion on the potential interaction modes that may dominate the stabilizing ability of Pt-tripod complexes.

Reviewer #3 (Remarks to the Author):

The paper titled "Solution structures of multiple G-quadruplex complexes regulated by a platinum(II)-based tripod reveal dynamic binding" by Mao describes two NMR structures of modified Tel26 human telomeric DNA in a complex with Pt-tripod ligand. One complex displays 1:1 stoichiometry with the ligand bound at 5'-end. Such coordination is observed in complexes of human telomeric DNA with a variety of other G-quadruplex (G4) ligands. The second complex is quite interesting, multimeric with 4:2 G4:Pt-tripod ratio. The work also includes a variety of biophysical methods (FL enhancement, Circular Dichroism titrations, PAGE, and MS) to further characterize the complexes studied as well as interaction of Pt-tripod ligand with other well known G4 structures (e.g. c-myc, bcl-2, etc). The experimental work is sound. Overall, the paper needs better writing, data interpretation, and presentation. In many cases, the interpretation of the data is overextended or oversimplified. Also, the authors do not generalize their findings – what did they learn from their work which could be used to design better G4 ligands as potential anticancer drugs?

My major concern is the appropriateness of the presented work for Nat. Commun. I am very delighted to see the structures of the ligand-G4 complexes, especially of the 4:2 complex. However, the Pt-tripod ligand with high positive charge is NOT specific for human telomeric DNA, it most likely binds equally well to duplex DNA and any other negatively charged polymer (See my comments below).

Major comments

2 Pt-tripod specificity. All throughout the paper the authors claim that Pt-tripod ligand binds selectively and specifically to wtTel26.

Here is one example: "Here, we find that the Pt-tripod selectively and specifically binds to the biological relevant hybrid-1 human telomeric G-quadruplex DNA Tel26". And just immediately down the page: "The results showed that Pt-tripod bound the hybrid-1 human telomeric G-quadruplex Tel26 (Fig. 1b) with much higher specificity and selectivity than other G-quadruplexes (Fig. 1c and Supplementary Fig. 1)"

When one examines FL enhancement data, gel electrophoresis, and FRET (Supplemental Figures 3, 5, and 6 respectively), Pt-tripod ligand behaves nearly identically in complex with ALL examined G4s. In fact, the complex between MYC and Pt-tripod moves as a single well defined band on PAGE, suggesting the formation of well folded single specie. In FRET, Pt-tripod stabilizes Bcl-2 to a

greater extent than Tel26.

I assume that the authors decided that Pt-tripod binding to other G4 is NOT specific, because 1H NMR spectra are broad. However, this could be explained not only by the aggregation caused by Pt-tripod ligand (this explanation is used by the authors), but also by intermediate time scale of Ligand-G4 tumbling.

Also, the authors did NOT demonstrate that Pt-tripod ligand does NOT bind to other DNA structures. So why do the authors claim that the binding is specific?

2. The notion of "Pt-tripod ... regulates unexpectedly" that is seen even in a title is unclear to me. I am very confused about the meaning of word "regulates" in this paper (e.g. "Pt-tripod gradually binds to Tel26 in stages, and induces and regulates monomeric, dimeric, and multimeric ... structures"). And also why higher order structures are unexpected. The ligand under investigation has +6 charge and I would not be surprised at all that it forms a higher order structure with a negatively charged DNA.

The figures (specifically 4 and 5) are so small that it is impossible to see the points discussed.

Minor comments

1. Fig. 2c: "After adding 2.0 equivalents of Pt-tripod, the major peak for the 2:1 Pt-tripod-Tel26 complex were detected (Fig. 2c)" I see in the Fig 2c the major peak for 1:1 and not for 2:1 complex.

2. The colors (yellow and red) in Fig. 2f are not clearly visible. Also, include the Pt-tripod complex alone on the gel.

3. Data interpretation is oversimplified, p9: "In addition, a new band with slower mobility was observed at approximately 35 bp, twice the size of the 1:1 complex. This band was fluorescent green and corresponded to the dimeric structure of the 4:2 complex, according to the ESI-MS data." The new band could suggest 4:2 complex, but could also be 3:2 complex or other higher order structure. Why 4:2 complex only is singled out?

4. 1:1 structure: based on Fig 3c, authors claim that the most structural changes come from the 5' end of the DNA: "Based on the complete assignment, the residues from the 5'-end exhibited the largest proton chemical shift changes", when in fact it seems that the chemical shifts of the bases 20-22 at the 3' end are also strongly affected by Pt-tripod binding. While the claims are not incorrect, they are definitely inaccurate or incomplete. Fig 4 c-e – very unclear; for example, H-bonding in Fig 4e is impossible to see.

5. 4:2 structure: unclear what does this sentence mean: "The TOCSY spectrum showed two sets of NOE connectivity from two bound Pt-tripods (Supplementary Fig. 14a), revealed that another Pt-tripod interacted with each Tel26 subunit in the dimeric complex." Also just next sentence: "Residues from the 3'-end displayed the largest proton chemical shift changes (Fig. 5c), indicating the 3'-end is the second binding site for the Pt-tripod." What about large chemical shift for residues in the middle, 12-16? Another statement: "As shown by the complex structure, the Pt-tripod interacts with the Tel26 not only through pi-pi stacking between the aromatic core and the external G-tetrads but also through electrostatic interactions of the cationic platinum groups (Fig. 6)." Which part of figure 6? I could not see any hint of the electrostatic interactions in this figure. No further details were provided in the text.

6. For both structures: the authors claim that "...the cationic platinum groups are positioned above the grooves and directed to the negative phosphate backbones of Tel26 for significant electrostatic interactions in a more dynamic conformation." I cannot see this neither in Fig 4 nor in Fig 5. Also what does it mean "in a more dynamic conformation"?

7. Can the authors overlay three structures (Tel26, 1:1 and 2:4 complex) to demonstrate better the observed conformational changes upon ligand binding.

8. What is the role of the third arm in the Pt-tripod ligand in binding to G4. Would it be better to prepare two-armed Pt complex based on your NMR structure?

9. The last statement of the Results seems to be too general and too unclear: "The unique dimeric structure provided useful structural information concerning the binding of biological relevant sequential G-quadruplex structures and the formation of binding pockets in the long telomere." ???

10. P 17; "We found the novel tertiary amine conformation ..." what exactly is novel about the conformation?

11. The data interpretation is overextended and/or oversimplified. I.e. "And we found the Pt-tripod cationic platinum groups are large enough to match G-quadruplex groove size such that they fit into the groove by electrostatic interactions, precisely fixing the position of the Pt-tripod." This statement does not have any experimental support and is based on pure speculation.

12. P17 Careless use of words: "totally nonplanar conformation"

13. P18 The statement "The induced structural change required to establish nice binding pockets in our structures are much different from the previously reported G-quadruplex-small molecule complex structures." How exactly different? What is different?

14. How was ligand concentration measured?

Point-by-point responses to the reviewers' comments

Dear Reviewers,

We thank you very much for your thoughtful comments and helpful suggestions for our manuscript. We have revised the manuscript based on your comments and our supplemental experiments. All of the revisions in the manuscript were highlighted in red. The point-by-point responses are listed below in blue.

The list of major revisions in the manuscript:

- (1). Second part of the result section: We added more explanation on the native PAGE and FRET results.
- (2). Third and fourth part of the result section: We added more explanation on the binding details of platinum units. In order to make it more clear, the structural details were also added in the new figure 4f-h and figure 6e-f.
- (3). Discussion section: We revised the discussion section.
- (4) New experiment data included in the manuscript:
 - a) More native PAGE experiments of wtTel26, bcl2Mid and MycG4 G-quadruplexes interacted with Pt-tripod. The gel pictures were visualized by the UV light (Supplementary Fig. 5).
 - b), The FRET melting assay of double-stranded DNA with Pt-tripod (Supplementary Fig. 6e).
 - c), The FRET competition experiment of Pt-tripod on human telomeric G-quadruplex (hTel) competed with different ratios of double-stranded calf thymus (CT) DNA (Supplementary Fig. 7).
- (5) We have made a video showing the binding of Pt-tripod and Tel26 (Supplementary Video 1).

The point-by-point responses to the reviewers' comments are as follows:

Reviewer #1 (Remarks to the Author):

The manuscript by Yang, Mao and their colleagues is focused on structural studies of DNA G-quadruplex interactions with a platinum(II) complex (Pt-tripod). A specific G-quadruplex studied originates from the biologically relevant hybrid-1 human telomeric repeat (Tel26) which has been implicated in inhibition of telomerase activity. Most excitingly, authors uncovered two modes of interactions and solved the 1:1 and the first dimeric 4:2 Pt-tripod-Tel26 complex structures by NMR. Structure calculations are supported with a large number of intermolecular NOEs between the Pt-tripod and Tel26. Scientific quality of this manuscript is high. Presentation and discussion of results are clear and supported with experimental data. Findings described in this manuscript are original and will be of interest to a wide range of scientific communities.

Response: Thank you for your kind comments.

As a comment, it could be suggested that authors try to discuss importance of new structures in view of promising properties of Pt-tripods as drugs.

Response: We have added discussion about the insights gained from the new structures in view of unique properties of Pt-tripods as drugs in the revised Discussion section. (Second and fourth paragraph, Pages 21, 22 and 23)

1. How is equilibrium between 1:1 and 4:2 complexes controlled?

Response: The equilibrium between 1:1 and 4:2 complexes were controlled by the stoichiometry of Pt-tripod/Tel26. As shown by the NMR, ESI-MS and native PAGE results (Fig. 1c, 2 and 4), the 1:1 complex was formed at 0.0-1.0 equivalents of Pt-tripod to Tel26. After adding 2.0 and 3.0 equivalents of Pt-tripod, the Pt-tripod continued interacting with Tel26 and induced the formation of 4:2 complex structure as shown by the NMR and native PAGE results (Fig. 2 and 6). Adding 4.0 and higher equivalents of Pt-tripod, multimeric complex structures were formed based on the native PAGE results (Fig. 2).

2. It is hard to follow the chemical shift differences in Figures 3c and 5c.

Response: We have enlarged and rearranged these two figures from one-column to two-column to make them clear.

Reviewer #2 (Remarks to the Author):

In this manuscript, authors studied a trinuclear Pt complex with a non-planar tripod structure and the interactions between the complex and several wild type and unnatural G-quadruplex (GQ) folding sequences with biological significance. The binding modes and the structural conformation of Pt-GQ complexes were studied under various stoichiometry of Pt/GQ. With the most stable binding to a revised telomeric GQ sequence, Tel26, the author identified two Pt-GQ with monomeric and dimeric Tel26 GQ. The novel structure of a 4:2 Pt-Tel complex was characterized by 1D and 2D NMR. This high order binding structure of Tel26 with Pt complex was also supported by ESI-MS spectra and native PAGE mobility analysis. This study offered breakthrough knowledge on the interactions between Pt complexes as anticancer drug candidates and various GQ topologies as drug targets on two novel levels. First, the proposed Pt complex possess a non-canonical structure for GQ binder. The complex has no extensive planar aromatic area for strong π - π stacking with G-quartets, which was normally considered to be the essential binding modes for all types of GQ binder to achieve high thermal and/or dynamic stability of GQs. Second, the manuscript offers, to my knowledge, the first NMR structure of high-order multimeric GQs with various binding stoichiometry of binder molecules. This structure would provide pivotal supports from structural biology to advance the studies in the biology and therapeutics of GQ structures and GQ targeting drug molecules. The experimental results and reference are sufficient and well presented to support author's arguments. I would like to recommend the publication of the manuscript after the following minor concerns were addressed by the authors.

Response: Thank you for your kind comments.

1. Figure S6 indicated that wtTel26 and Tel26 both showed a much higher thermal melting temperatures under high stoichiometric ratio of Pt vs GQ. The authors attribute the results as an evidence of the existence of high-order structures. However, the melting curves showed clearly two transitions in UV absorption intensity. The authors need to discuss in more details about the melting process. What is the structural transition occurring at these two transitions, respectively? Are the two transition a two step melting of the high-order structure or melting processes of two individual types of folding structures, one low-order monomeric structure and one high-order dimeric GQ structure? The thorough discussion on thermal melting curve would offer more information of folding/unfolding dynamics of the high-order Pt-GQ complexes.

Response: We have added a detailed discussion of the thermal melting data in the revised manuscript. (Page 9)

The structural transitions showing in the FRET results are the melting processes of two individual types of folding structures, one monomeric complex structure and one 4:2 dimeric G-quadruplex complex structure, according to our other results including NMR and native PAGE. At a 1:1 ratio of Pt-tripod/Tel26, there is one melting point (~55 °C) showing in the slope plot, corresponding to the 1:1 complex (Supplementary Fig. 6a). At a 2:1 and 3:1 ratio of Pt-tripod/Tel26, the slope plot exhibited two melting transitions, revealing the melting processes of two individual structures. The first melting point was belonging to the 1:1 complex as shown by the same ΔT_m . The second melting point with a huge ΔT_m , belonging to the stable structure of the 4:2 complex. The coexistence of the two complexes is shown in the native PAGE data (Fig. 2e,f).

2. The manuscript described the NMR structures of 1:1 and 4:2 Pt-Tel26 GQ complexes. In discussion part, authors also claim that the three Pt cationic cores can fit into the grooves and lock the complex to GQ monomer and dimers via electrostatic attraction. However, in result part, the structural information of Pt cores in either 1:1 or 4:2 Pt-Tel26 complexes were mentioned. The claims that tripod Pt complex offer the unique interaction modes via Pt cationic cores with GQ grooves in the discussion part seems quite out of sudden. More result explanation on this issue would significantly strengthen the manuscript.

Response: Thanks for raising this important question. We have added more detailed structural information and explanation of the interactions of Pt cationic cores with GQ grooves in the Results section in the revised manuscript. (for 1:1 complex, page 14; for 4:2 complex, page 18)

In order to make it more clear, we have added structural details in the new Fig. 4f-h and Fig. 6e-f, showing hydrogen bonding and electrostatic interactions of platinum units in the 1:1 and 4:2 Pt-tripod-Tel26 complexes.

3. As the author briefly mentioned in the discussion, “Compounds with a more flexible central core are unlikely to strongly bind to G-quadruplexes. However, in this work, the Pt-tripod is a tertiary amine with a totally nonplanar conformation. Nevertheless, it specifically recognizes the hybrid-1 human telomeric G-quadruplex.” The unique stable high-order structures were achieved by Pt-tripod complexes without the classic π - π stacking. The phenomenon is very inspiring to the design of GQ-targeting compounds and may open the structure library significantly. However, the discussion on elaborating the novel interaction modes offered by Pt-tripod complexes was quite limited in the current manuscript. To enhance the impact of the manuscript, I suggest the authors

provide more thorough discussion on the potential interaction modes that may dominate the stabilizing ability of Pt-tripod complexes.

Response: We have added more detailed discussion about the potential interaction modes between the Pt-tripod and Tel26 that contribute to the stabilizing ability of Pt-tripod complexes. (Second and fourth paragraph of the revised Discussion, Pages 21, 22 and 23)

Reviewer #3 (Remarks to the Author):

The paper titled “Solution structures of multiple G-quadruplex complexes regulated by a platinum(II)-based tripod reveal dynamic binding” by Mao describes two NMR structures of modified Tel26 human telomeric DNA in a complex with Pt-tripod ligand. One complex displays 1:1 stoichiometry with the ligand bound at 5'-end. Such coordination is observed in complexes of human telomeric DNA with a variety of other G-quadruplex (G4) ligands. The second complex is quite interesting, multimeric with 4:2 G4:Pt-tripod ratio. The work also includes a variety of biophysical methods (FL enhancement, Circular Dichroism titrations, PAGE, and MS) to further characterize the complexes studied as well as interaction of Pt-tripod ligand with other well known G4 structures (e.g. c-myc, bcl-2, etc). The experimental work is sound.

Response: Thank you for your kind comments.

Overall, the paper needs better writing, data interpretation, and presentation. In many cases, the interpretation of the data is overextended or oversimplified.

Response: We have extensively revised the manuscript to improve clarity and data interpretation/presentation.

Also, the authors do not generalize their findings – what did they learn from their work which could be used to design better G4 ligands as potential anticancer drugs?

Response: We have extensively revised and extended the Discussion section to provide detailed information about the unique interactions of the Pt-tripod in the two complexes, as well as the relevance to the designing of G4 ligands. (Second and fourth paragraph of the revised Discussion, Pages 21, 22 and 23)

My major concern is the appropriateness of the presented work for Nat. Commun. I am very delighted to see the structures of the ligand-G4 complexes, especially of the 4:2 complex. However, the Pt-tripod ligand with high positive charge is NOT specific for human telomeric DNA, it most likely binds equally well to duplex DNA and any other negatively charged polymer (See my comments below).

Major comments

1. Pt-tripod specificity. All throughout the paper the authors claim that Pt-tripod ligand binds selectively and specifically to Tel26. Here is one example: “Here, we find that the Pt-tripod selectively and specifically binds to the biological relevant hybrid-1 human telomeric G-quadruplex DNA Tel26”. And just immediately down the page: “The results showed that Pt-tripod bound the hybrid-1 human telomeric G-quadruplex Tel26 (Fig. 1b) with much higher specificity and selectivity than other G-quadruplexes (Fig. 1c and Supplementary Fig. 1)”.

When one examines FL enhancement data, gel electrophoresis, and FRET (Supplemental Figures 3, 5, and 6 respectively), Pt-tripod ligand behaves nearly identically in complex with ALL examined G4s. In fact, the complex between MYC and Pt-tripod moves as a single well defined band on PAGE, suggesting the formation of well folded single specie. In FRET, Pt-tripod stabilizes Bcl-2 to a greater extent than Tel26.

I assume that the authors decided that Pt-tripod binding to other G4 is NOT specific, because ¹H NMR spectra are broad. However, this could be explain not only by the aggregation caused by Pt-tripod ligand (this explanation is used by the authors), but also by intermediate time scale of Ligand-G4 tumbling.

Also, the authors did NOT demonstrate that Pt-tripod ligand does NOT bind to other DNA structures. So why do the authors claim that the binding is specific?

Response: Thanks for this important comment.

Also, the authors did NOT demonstrate that Pt-tripod ligand does NOT bind to other DNA structures. So why do the authors claim that the binding is specific?

We have performed experiments on the binding of Pt-tripod with dsDNA, FRET melting (Supplementary Fig. 6e), and especially FRET competition experiments (Supplementary Fig. 7) with the 1:1 Pt-tripod-Tel26 complex. The results show that Pt-tripod binds much stronger to Tel26 than dsDNA. (Pages 9, 10)

1. Pt-tripod specificity. All throughout the paper the authors claim that Pt-tripod ligand binds selectively and specifically to Tel26. Here is one example: “Here, we find that the Pt-tripod selectively and specifically binds to the biological relevant hybrid-1 human telomeric G-quadruplex DNA Tel26”. And just immediately down the page: “The results showed that Pt-tripod bound the hybrid-1 human telomeric G-quadruplex Tel26 (Fig. 1b) with much higher specificity and selectivity than other G-quadruplexes (Fig. 1c and Supplementary Fig. 1)”.

I assume that the authors decided that Pt-tripod binding to other G4 is NOT specific, because ^1H NMR spectra are broad. However, this could be explain not only by the aggregation caused by Pt-tripod ligand (this explanation is used by the authors), but also by intermediate time scale of Ligand-G4 tumbling.

We agree it is inaccurate to state that “Pt-tripod binds selectively to Tel26”, as Pt-tripod is shown to bind other G4s. We have removed all the related statements in the revised manuscript. For “The specific binding of Pt-tripod to hybrid-1 Tel26”, we mean that Pt-tripod binds hybrid-1 Tel26 to form well-defined complex(es). We have clarified this in the revised manuscript. The specific binding of Pt-tripod to hybrid-1 Tel26 is shown by the NMR titration and 2D data, as well as the native PAGE data. The NMR titration data clearly shows that Pt-tripod only forms well-defined complex structure(s) when binding the Tel26. (Actually, Pt-tripod also appears to form a 1:1 complex with the hybrid-2 wtTel26, but not the dimeric complex. We have added this point in the revised manuscript). The native PAGE data show that clear 1:1 and dimeric complexes are formed with Tel26, but not other G4 sequences.

Yes, we decided that Pt-tripod binding to Tel26 specifically based on the well-defined ^1H NMR spectra. We are not sure if we understand “...but also by intermediate time scale of Ligand-G4 tumbling” completely. The broad NMR spectra can be caused by Ligand-G4 tumbling, however, different G4s (including Tel26) and G4-ligand complexes (of the same stoichiometry) are very similar in size and thus would have similar tumbling rates, unless the complexes are of much higher molecularity (higher-order) and thus much greater molecular weights. Or this is about the intermediate exchange time scale of ligand binding, which would affect the NMR linewidth but is closely related to binding affinity.

When one examines FL enhancement data, gel electrophoresis, and FRET (Supplemental Figures 3, 5, and 6 respectively), Pt-tripod ligand behaves nearly identically in complex with ALL examined G4s. In fact, the complex between MYC and Pt-tripod moves as a single well defined band on PAGE, suggesting the formation of well folded single specie. In FRET, Pt-tripod stabilizes Bcl-2 to a greater extent than Tel26.

We have performed more native PAGE experiments (Supplementary Fig. 5).

As shown in the gel, for MYC G4, even *at 1:1 ratio* of Pt-tripod, the single species corresponds to the higher-order structure (~50 bp, similar to the multimeric structures (> 4:2 complex) observed for Tel26 at ratios >4:1) (Supplementary Fig. 5c), therefore, no 1:1 complex is formed with Myc G4. For Bcl-2 G4, *at 1:1 ratio*, three bands were observed in the gel, corresponding to monomeric free DNA (fluorescence red, below 15 bp control), Pt-tripod-complex 1 (~22 bp control, thus not monomeric, likely dimeric), Pt-tripod-complex 2 (higher-order, 30 bp control) (Supplementary Fig. 5b). Therefore, Bcl-2 G4 also doesn't form a clear 1:1 complex with Pt-tripod; the higher FRET T_m corresponds to dimeric and higher-order structures. Actually, the T_m values of all G4 higher-order structures are similar (Supplementary Fig. 6).

We have added these new data in the revised SI. We have also added detailed data interpretation for FRET melting in the revised manuscript text (Pages 9, 10).

2. The notion of “Pt-tripod ... regulates unexpectedly” that is seen even in a title is unclear to me. I am very confused about the meaning of word “regulates” in this paper (e.g. “Pt-tripod gradually binds to Tel26 in stages, and induces and regulates monomeric, dimeric, and multimeric ... structures”).

Response: We agree with the reviewer and we have changed the word “regulate” to “induce”.

And also why higher order structures are unexpected. The ligand under investigation has +6 charge and I would am not surprised at all that it forms a higher order structures with a negatively charged DNA.

Response: We have removed “unexpected” for the higher order structures in the revised manuscript.

The figures (specifically 4 and 5) are so small that it is impossible to see the points discussed.

Response: We have enlarged and revised these two figures to make them clearer.

Minor comments

1. Fig. 2c: “After adding 2.0 equivalents of Pt-tripod, the major peak for the 2:1 Pt-tripod–Tel26 complex were detected (Fig. 2c)” I see in the Fig 2c the major peak for 1:1 and not for 2:1 complex.

Response: We have revised it in the manuscript on page 7, as shown below:

“After adding 2.0 equivalents of Pt-tripod, the major peaks for 1:1 and 2:1 Pt-tripod–Tel26 complexes were detected (Fig. 2c and Supplementary Fig. 4c).”

2. The colors (yellow and red) in Fig. 2f are not clearly visible. Also, include the Pt-tripod complex alone on the gel.

Response: We have performed this experiment again including the free Pt-tripod complex alone on the gel and made new figures as suggested (Fig. 2e,f).

3. Data interpretation is oversimplified, p9 : “In addition, a new band with slower mobility was observed at approximately 35 bp, twice the size of the 1:1 complex. This band was fluorescent green and corresponded to the dimeric structure of the 4:2 complex, according to the ESI-MS data.” The new band could suggest 4:2 complex, but could also be 3:2 complex or other higher order structure. Why 4:2 complex only is singled out?

Response: The assignment of the band to the 4:2 complex is based on the NMR titration data and NMR structures in solution. We have provided an explanation for the observation of the 3:2 complex in ESI-MS. (Page 7)

We have revised this part of our manuscript on page 8 as below:

“At 2.0 Pt-tripod equivalents, a new band fluorescent green with slower mobility was observed at approximately 35 bp, corresponding to a dimeric structure.”

4. 1:1 structure: based on Fig 3c, authors claim that the most structural changes come from the 5' end of the DNA: “Based on the complete assignment, the residues from the 5'-end exhibited the largest proton chemical shift changes”, when in fact it seems that the chemical shifts of the bases 20-22 at the 3' end are also strongly affected by Pt-tripod binding. While the claims are not incorrect, they are definitely inaccurate or incomplete.

Response: We defined the 5'-end and 3'-end residues based on the G-quadruplex structure. As shown in the Tel26 G-quadruplex structure (Fig. 1b), the residues 20-22 are

at the 5' G-tetrad or 5'-loop region of the G-quadruplex structure. We have revised our manuscript text to be more accurate. (Page 12)

“Based on the complete assignment, the residues from the 5' G-tetrad (G4, G10, G18 and G22) exhibited the largest proton chemical shift changes (Fig. 3c), indicating Pt-tripod stacking at the 5' G-tetrad. Notably, large proton chemical shift changes were also observed for 5' loop and flanking regions (A1, A2, A3, T8, A9, T20 and A21), which revealed important conformational rearrangements and ligand interactions at the Pt-tripod binding site at the 5'-end of Tel26.”

Fig 4 c-e – very unclear; for example, H-bonding in Fig 4e is impossible to see.

Response: We have enlarged and revised the Fig. 4 to make it more clear. And we have added structural details in the new Fig. 4f-h, showing hydrogen bonding and electrostatic interactions of Pt groups in the 1:1 Pt-tripod-Tel26 complex.

5. 4:2 structure: unclear what does this sentence mean: “The TOCSY spectrum showed two sets of NOE connectivity from two bound Pt-tripods (Supplementary Fig. 14a), revealed that another Pt-tripod interacted with each Tel26 subunit in the dimeric complex.”

Response: Thanks for pointing this misstatement out. We have revised this sentence. (Page 17)

“The binding of the second Pt-tripod is supported by the TOCSY and NOESY spectra, which showed two sets of crosspeaks from two bound Pt-tripods (Supplementary Fig. 15), revealing another Pt-tripod interacted with the 3' of Tel26 G-quadruplex.”

Also just next sentence: “Residues from the 3'-end displayed the largest proton chemical shift changes (Fig. 5c), indicating the 3'-end is the second binding site for the Pt-tripod.”
What about large chemical shift for residues in the middle, 12-16?

Response: We defined the 5'-end and 3'-end residues based on the G-quadruplex structure. As shown in the Tel26 G-quadruplex structure (Fig. 1b), the residues 12-16 are at the 3' G-tetrad or 3'-loop region of the G-quadruplex structure.

We have revised our manuscript to be more accurate and to include more detailed explanation. (Page 17)

“Residues from the 3' G-tetrad (G6, G12, G16 and G24) and 3' loop and flanking regions (T13, T14, A15, A25 and A26) displayed the largest proton chemical shift changes (Fig. 5c), indicating the 3'-end is the second binding site for the Pt-tripod and that the binding of Pt-tripod causes a large conformation reorientation in the 3'-capping regions.”

Another statement: “As shown by the complex structure, the Pt-tripod interacts with the Tel26 not only through pi-pi stacking between the aromatic core and the external G-tetrads but also through electrostatic interactions of the cationic platinum groups (Fig. 6).” Which part of figure 6? I could not see any hint of the electrostatic interactions in this figure. No further details were provided in the text.

Response: We have added structural details in the new Fig. 6e-f, showing potential hydrogen bonding and electrostatic interactions of platinum groups in the 4:2 Pt-tripod-Tel26 complex. We have also provided more details on the electrostatic interactions of the Pt groups in the revised manuscript text. (Page 18)

6. For both structures: the authors claim that “...the cationic platinum groups are positioned above the grooves and directed to the negative phosphate backbones of Tel26 for significant electrostatic interactions in a more dynamic conformation.” I cannot see this neither in Fig 4 no in Fig 5.

Response: We have added structural details in the new Fig. 4f-h and Fig. 6e-f, showing hydrogen bonding and electrostatic interactions of platinum units with the negative phosphate backbones or the loop residues in the 1:1 and 4:2 Pt-tripod-Tel26 complexes. And we have also added more explanation on them in the revised manuscript text. (Pages 14 and 18)

Also what does it mean “in a more dynamic conformation”?

Response: We have revised this statement.

7. Can the authors overlay three structures (Tel26, 1:1 and 2:4 complex) to demonstrate better the observed conformational changes upon ligand binding?

Response: We have added a new figure showing the overlay of the three structures (Supplementary Fig. 17).

8. What is the role of the third arm in the Pt-tripod ligand in binding to G4.

Response: We have added the description of the role of the third arm in the Pt-tripod ligand binding to Tel26 in the revised manuscript. (Pages 14 and 18)

Would it be better to prepare two-armed Pt complex based on your NMR structure?

As shown in our Discussion section on pages 21, 22 and 23, the stabilization of loop and flanking regions by the third arm of Pt-tripod is important for the stabilization of the whole complex structure and further formation of higher-order G-quadruplex structures. Two-armed Pt compound would not stabilize most of the loop and flanking regions in the complex structure and may result in formation of non-well-defined complex structures.

9. The last statement of the Results seems to be too general and too unclear: “ The unique dimeric structure provided useful structural information concerning the binding of biological relevant sequential G-quadruplex structures and the formation of binding pockets in the long telomere.” ?

Response: We have deleted this statement.

10. P 17; “We found the novel tertiary amine conformation ...” what exactly is novel about the conformation?

Response: We have revised this sentence to be more clear and accurate. We have also included more detailed discussion about this point in the revised discussion. (Page 23)

11. The data interpretation is overextended and/or oversimplified. I.e. “And we found the Pt-tripod cationic platinum groups are large enough to match G-quadruplex groove size such that they fit into the groove by electrostatic interactions, precisely fixing the position of the Pt-tripod.” This statement does not have any experimental support and is based on pure speculation.

Response: We agree and we have deleted this statement. Instead, we have included more detailed discussion based on experimental data in the revised Discussion section. (Pages 21-23)

12. P17 Careless use of words: “totally nonplanar conformation”

Response: We have deleted the word “totally”.

13. P18 The statement “The induced structural change required to establish nice binding pockets in our structures are much different from the previously reported G-quadruplex-small molecule complex structures.” How exactly different? What is different?

Response: We have included more detailed discussion about this point in the revised discussion. (First and second paragraph of the Discussion section, Pages 21 and 22)

14. How was ligand concentration measured?

Response: The pure Pt-tripod powder was characterized by NMR, ESI-MS and elemental analysis to obtain the chemical formula of the Pt-tripod. We made the stock solution of Pt-tripod based on mass measurement, so the ligand concentration was determined based on mass.

REVIEWERS' COMMENTS:

Reviewer #1 (Remarks to the Author):

Authors have responded to all the comments in satisfactory fashion. Manuscript is ready to be published.

Reviewer #2 (Remarks to the Author):

The authors have revised the manuscript, especially the decision part to address my questions and comments. I believe the current version is well deserved to be published in Nature Comm and will attract and inspire a wide spectrum of readership.

Reviewer #3 (Remarks to the Author):

Authors have done good experimental work and significant rewriting, both of which improved this manuscript.

The reviewer, however, still has few minor points.

1. The reviewer was delighted to see FREE binding experiments with dsDNA as well as FRET competition experiments with CT DNA. However, the reviewer could not find any information about how CT DNA concentration was measured. The reviewer thus assumes that, as usually done, the extinction coefficient for CT DNA is provided per base pair. In that case the statement:

“the stability of 1:1 Pt-tripod–Tel26 complex remains 81% and 51% after adding 10 and 100 folds of calf thymus (CT) DNA, respectively, illustrating the Pt-tripod has much higher binding affinity with human telomeric G-quadruplex DNA over double-stranded DNA”

is not that impressive. It is not ‘much higher’ stability as 100 fold CT concentration is only 100 bp fold and probably mole like only 10-fold better binding to GQ DNA vs dsDNA, which is good but not impressive and most likely not sufficient for any reasonable biological application as dsDNA is significantly more abundant as compared to GQ DNA. In our own work we usually use up to 480 fold excess of CT in FRET competitions.

2. For SI figure 17, provide rmsds between the structures to complement the figures.

3. The authors state that ‘the 3’ binding pocket is better-defined’ yet ligand prefers 5’binding pocket. Does the structure provides any explanation for this observation?

4. Discussion needs to be rewritten for clarity and shorten

Point-by-point responses to the reviewers' comments

Dear Reviewers,

We thank you very much for your thoughtful comments and helpful suggestions for our manuscript. We have revised the manuscript based on your comments. All of the revisions in the manuscript were using the 'track changes' feature of Microsoft Word. The point-by-point responses are listed below in blue.

Reviewer #1 (Remarks to the Author):

Authors have responded to all the comments in satisfactory fashion. Manuscript is ready to be published.

Reviewer #2 (Remarks to the Author):

The authors have revised the manuscript, especially the decision part to address my questions and comments. I believe the current version is well deserved to be published in Nature Comm and will attract and inspire a wide spectrum of readership.

Reviewer #3 (Remarks to the Author):

Authors have done good experimental work and significant rewriting, both of which improved this manuscript.

The reviewer, however, still has few minor points.

1. The reviewer was delighted to see FREE binding experiments with dsDNA as well as FRET competition experiments with CT DNA. However, the reviewer could not find any information about how CT DNA concentration was measured. The reviewer thus assumes that, as usually done, the extinction coefficient for CT DNA is provided per base pair. In that case the statement:

“the stability of 1:1 Pt-tripod–Tel26 complex remains 81% and 51% after adding 10 and 100 folds of calf thymus (CT) DNA, respectively, illustrating the Pt-tripod has much higher binding affinity with human telomeric G-quadruplex DNA over double-stranded DNA”

is not that impressive. It is not ‘much higher’ stability as 100 fold CT concentration is only 100 bp fold and probably mole like only 10-fold better binding to GQ DNA vs dsDNA, which is good but not impressive and most likely not sufficient for any reasonable biological application as dsDNA is significantly more abundant as compared to GQ DNA. In our own work we usually use up to 480 fold excess of CT in FRET competitions.

Response: The concentration of CT DNA is reported per base pair and we have added illustration on page 10.

We have added explanations about how CT DNA concentration was measured in the Methods section (page 28).

And we have read the related literature and cited it in our manuscript (reference 48). We have changed “much higher” into “higher” in the explanation, which is shown as:

“the stability of 1:1 Pt-tripod–Tel26 complex remains 81% and 51% after adding 10 and 100 folds (base pair) of calf thymus (CT) DNA, respectively, illustrating the Pt-tripod has higher binding affinity with human telomeric G-quadruplex DNA over double-stranded DNA”

2. For SI figure 17, provide rmsds between the structures to complement the figures.

Response: We have added rmsds for the overlapped structures (Supplementary Figure 25).

Because of the format editing, the numbering of supplementary figures were changed. Supplementary figure 17 changed into supplementary figure 25.

3. The authors state that ‘the 3’ binding pocket is better-defined’ yet ligand prefers 5’binding pocket. Does the structure provide any explanation for this observation?

Response: Compared with the structural details of the 5’ and 3’ binding pockets, we found the non-hydrogen-bonded A3-A9-T20 capping triad at the 5’-end is more flexible. The loop residue T8 moved close to interact with the third platinum unit. As a result, the residue A9 stayed away from T20 and it is hard to form hydrogen bonding between A3, T20 and A9 (Fig. 4d). Moreover, the 5’-flanking residues A1 and A2 are flexible and exposed to solvent.

At the 3’-end, the capping residues T13, T14 and A25 did not interact with platinum units, so it is easier for them to move close and form hydrogen bonding between them. The

hydrogen bonded T13:A25:T14 capping triad is further stacked and stabilized by the A26:A26* pair (Fig. 6d,g).

So we claimed that, the 3' binding pocket is better-defined than the 5' binding pocket.

4. Discussion needs to be rewritten for clarity and shorten.

Response: Thank you for your advice. We have revised and shortened the discussion section to make it clearer.